# Random forests with spatial proxies for environmental modelling: opportunities and pitfalls

Carles Milà[1,2], Marvin Ludwig[3], Edzer Pebesma[4], Cathryn Tonne[1,2,5], and Hanna Meyer[3]

[1]Barcelona Institute for Global Health (ISGlobal), Barcelona, Spain
[2]Universitat Pompeu Fabra (UPF), Barcelona, Spain
[3]Institute of Landscape Ecology, University of Münster, Münster, Germany
[4]Institute of Geoinformatics, University of Münster, Münster, Germany
[5]CIBER epidemiología y salud pública (CIBERESP), Madrid, Spain

**Correspondence:** Carles Milà (carles.mila@isglobal.org)

**Abstract.** Spatial proxies such as coordinates and distance fields are often added as predictors in Random Forest (RF) models without any modification of the algorithm to account for residual autocorrelation and improve predictions; however, their suitability under different predictive conditions encountered in environmental applications has not yet been assessed. We investigate 1) the suitability of spatial proxies depending on the modelling objective (interpolation vs. extrapolation), the strength of the residual spatial autocorrelation, and the sampling pattern; 2) which validation methods can be used as a model selection tool to empirically assess the suitability of spatial proxies; and show 3) the effect of using spatial proxies in real-world environmental applications.

We designed a simulation study to assess the suitability of RF regression models using three different types of spatial proxies: coordinates, Euclidean Distance Fields (EDF), and Random Forest spatial prediction (RFsp). We also tested the ability of probability sampling test points, random k-fold Cross-Validation (CV), and k-fold Nearest Neighbour Distance Matching (kNNDM) CV to reflect the true prediction performance and correctly rank models. As real-world study cases, we modelled annual average air temperature and fine particulate matter air pollution for continental Spain.

In the simulation study, we found that RF with spatial proxies was poorly suited for spatial extrapolation to new areas due to large feature extrapolation. For spatial interpolation, proxies were beneficial when both strong residual autocorrelation, and regularly or randomly-distributed training samples, were present. In all other cases, proxies were neutral or counterproductive. Random k-fold cross-validation generally favoured models with spatial proxies even when not appropriate, whereas probability test samples and kNNDM CV correctly ranked models. In the study cases, air temperature stations were well-spread within the prediction area and measurements exhibited strong spatial autocorrelation, leading to an effective use of spatial proxies. Air pollution stations were clustered and autocorrelation was weaker, and thus spatial proxies were not beneficial.

As the benefits of spatial proxies are not universal, we recommend using spatial exploratory and validation analyses to determine their suitability, as well as considering alternative inherently spatial modelling approaches.

# 1 Introduction

Predictive modelling of environmental data is key to produce spatially-continuous information from limited, typically expensive and hard-to-collect point samples. Research fields as diverse as meteorology (Kloog et al., 2017), soil sciences (Poggio et al., 2021), ecology (Ma et al., 2021), and environmental epidemiology (de Hoogh et al., 2018) rely on predictive mapping workflows to produce continuous surfaces, sometimes even at global scale (Ludwig et al., 2023), with products being used for decision-making and subsequent modelling.

Spatial data including environmental variables have intrinsic characteristics that impact the way they are modelled (Longley, 2005). One of the most important is spatial autocorrelation, which modellers have used to support their spatial interpolation endeavours that evolved from deterministic univariate approaches such as inverse distance weighting, to more advanced geostatistical methods that leverage auxiliary predictor information such as regression kriging (Heuvelink and Webster, 2022). With the increasing availability of spatial data relevant to predict environmental variables (e.g. new satellites and sensors, climatic and atmospheric simulations), Machine Learning (ML) models have gained momentum due to their ability to capture complex non-linear relationships in highly dimensional datasets (Lary et al., 2016). While standard ML models can better capture complexity in the trend estimation compared to regression kriging, they are *aspatial*, i.e. they ignore the spatial location of the samples and assume independence between observations (Wadoux et al., 2020a). One of the most popular ML algorithms in the geospatial community is Random Forest (RF), a decision tree ensemble (Breiman, 2001) that has shown good performance across many applications (Wylie et al., 2019) and centred the attention of many methodological studies (e.g. Meyer and Pebesma, 2021; Hengl et al., 2018; Sekulić et al., 2020; Georganos et al., 2021; Saha et al., 2023).

The lack of consideration of space in ML models has motivated researchers to try to find ways to account for spatial autocorrelation to improve model performance. One straightforward approach is to add "spatial proxies" as predictors to the ML model without any modification of the algorithm. We define spatial proxies as a set of spatially-indexed variables with long or infinite autocorrelation ranges that are not causally related to the response. We use the term "proxy" since these predictors act as surrogates for unobserved factors, such as missing predictors or an autocorrelated error term, that can cause residual autocorrelation. The most prevalent type of proxy are coordinates, where either geographical or projected coordinate fields (Fig. 1.3) are added as two additional predictors in the models (e.g. Cracknell and Reading, 2014). Other spatial proxy approaches include Euclidean Distance Fields (EDF) (Behrens et al., 2018) which, in addition to coordinates, adds additional distance fields with different origins, such as five EDF with respect to the four corners and the centre of the study area (Fig. 1.3). Behrens et al. (2018) explained that with EDF one can account for both spatial autocorrelation and non-stationarity by using the partition of the geographical space introduced by EDF and its interaction with the environmental predictors. Finally, Hengl et al. (2018) proposed Random Forest spatial prediction (RFsp), which adds distance fields to each of the sampling locations (Fig. 1.3), i.e. the number of added predictors equals the sample size. Hengl et al. (2018) argued that RFsp can address spatial autocorrelation, model trend and error in a single step, mimick regression kriging while avoiding its complexity and assumptions, and benefit from the ability of RF to fit complex relationships between the response and predictors.

While spatial proxies, and especially coordinates, have been widely used in the literature (e.g. Walsh et al., 2017; Wang et al., 2017; de Hoogh et al., 2018), the evidence exploring their suitability in different prediction settings is fragmented and limited. In our literature review, we identified three factors that could affect the effectiveness of spatial proxies: 1) the models' objective, 2) the strength of the residual spatial autocorrelation, and 3) the sample distribution.

In relation to the first factor, the objective of the model, we can distinguish: interpolation, where there is a geographical overlap between the sampling and prediction area; extrapolation or spatial model transfer, where the model is applied to a new, disjoint area; and predictive inference, where knowledge discovery is the main focus. Regarding interpolation, several studies indicate that, when samples cover the entire prediction area, the addition of spatial proxies to RF may be beneficial in terms of predictive accuracy and might outperform geostatistical or hybrid methods (Behrens et al., 2018; Hengl et al., 2018; Saha et al., 2023). The use of spatial proxies for extrapolation remains to be explored but appears to be problematic: since the spatial representation is introduced via predictors, and the prediction area is, by definition, different than the sampling area, feature extrapolation will be present when spatial proxies are used, which is problematic for models with poor extrapolation ability such as RF (Meyer and Pebesma, 2021; Hengl et al., 2018). Finally, regarding predictive inference, the inclusion of spatial proxies has been discouraged: Meyer et al. (2019) showed how spatial proxies typically rank high in variable importance statistics in RF models, especially when they lead to overfitting. Following this, Wadoux et al. (2020a) discussed how high proxy variable importance could hinder correct interpretation of importance statistics for the rest of predictors, which could undermine the possibility to derive hypotheses from the model and hamper residual analysis.

The second factor is residual autocorrelation, which typically arises when a relevant predictor is not available for modelling because it is either unmeasured or unknown, or because the error term is autocorrelated (F. Dormann et al., 2007). Since the goal of introducing spatial proxies is to account for residual autocorrelation, a better performance of models with spatial proxies is expected when residual dependencies are strong. This intuition is confirmed by the results of Saha et al. (2023), who showed how RF with spatial proxies, and especially those adding a large number of proxy predictors such as RFsp, were especially useful when the covariate signal to spatial noise ratio was low (i.e. large autocorrelated error term compared to the covariate signal), yet led to poor results when the spatial error was small. Nonetheless, whether proxies can address different sources of residual autocorrelation, i.e. missing predictors or autocorrelated error, as well as the influence of the strength of their spatial structure, remains to be studied.

The third factor is the sampling pattern, with clustered samples frequently argued to be potentially problematic (Cracknell and Reading, 2014; Hengl et al., 2018; Meyer et al., 2019). Indeed, the problem with clustered data is similar to that of spatial model transferability: even if the sampling and the prediction area coincide, there will be some regions not covered by the training data and therefore spatial extrapolation will occur to some degree. Cracknell and Reading (2014) showed that using coordinates with clustered data led to unplausible results with significant artifacts. Hengl et al. (2018) warned about using RFsp with clustered data which can result in feature extrapolation for a subset of the area, i.e. predicting for values of spatial proxies not included in the training data. Meyer et al. (2019) added that including highly autocorrelated variables such as coordinates with clustered samples can result in spatial overfitting. In spite of this evidence, the effect of the sampling design has only been explored through specific study cases and a systematic evaluation is still missing.

In addition to the factors influencing the suitability of spatial proxies, it is important to have validation methods to empirically assess whether a spatial proxy approach is advisable in a given prediction task. To our knowledge, the only evidence regarding this point is that of Meyer et al. (2019), who showed that spatial overfitting with highly autocorrelated variables was only detected when using an appropriate validation strategy. Amongst validation methods, probability test sampling is the preferred approach as it offers unbiased estimates (Wadoux et al., 2021) that can be used for model selection. Unfortunately, independent test samples are rarely available in the field of environmental sciences, and alternative validation methods such as Cross-Validation (CV) must be used. While standard CV methods that assume independence between train and test data such as leave-one-out and k-fold CV have been acknowledged to offer good accuracy estimates for spatial interpolation with regular and random samples (Wadoux et al., 2021; Milà et al., 2022; Linnenbrink et al., 2023), they generally lead to overoptimistic estimates for spatial model transfer and interpolation with clustered samples. Several spatial CV methods have been proposed to address the limitations of standard validation approaches (Roberts et al., 2017; Ploton et al., 2020; Kattenborn et al., 2022) using CV based on spatial blocking (Wenger and Olden, 2012; Valavi et al., 2019), buffering (Telford and Birks, 2009; Le Rest et al., 2014), clustering (Wang et al., 2023), as well as sampling-intensity weighted CV and model-based geostatistical approaches (de Bruin et al., 2022). Among those, CV methods that consider the prediction objective of the model such as k-fold Nearest Neighbour Distance Matching (kNNDM) (Linnenbrink et al., 2023) are especially interesting because they have the potential to discern whether proxies are useful for different prediction objectives, i.e. interpolation vs. extrapolation.

As an alternative to modelling with spatial proxies, other methods that *do* involve algorithmic modifications have been proposed, including mixed effects tree-based models that account for correlated data (Hajjem et al., 2011, 2014), spatially-aware resampling methods (Li et al., 2019), as well as geographically weighted ML algorithms (Georganos et al., 2021; Zhan et al., 2017). Among those, the Random Forest-Generalized Least Squares (RF-GLS) model recently proposed by Saha et al. (2023) is especially interesting because it relaxes the independence assumption of the RF model by accounting for spatial dependencies in several ways: 1) a global dependence split criterion and node representatives instead of the CART criterion used in standard RF models; 2) contrast resampling rather than bootstrap used in standard RF; 3) residual kriging with covariance modelled using a Gaussian process framework (Saha et al., 2023). In their simulations, Saha et al. (2023) showed how RF-GLS outperformed RF with and without spatial proxies; however, their simulations did not reflect the typical characteristics of environmental applications as they only explored random sampling designs and did not use spatially-structured predictors.

Even though their strengths and weaknesses have been discussed, spatial proxies continue to be widely used and coordinates are typically added to the set of predictors by default without further consideration. Hence, a comprehensive investigation is required to complement the fragmented evidence, mostly available from study cases, that is currently available. In this work, we investigate several RF models with spatial proxies, namely coordinates, EDF, and RFsp, with the following objectives:

1. To assess the suitability of spatial proxies depending on different factors: the modelling objective (interpolation vs. extrapolation), the strength of the residual spatial autocorrelation, and the sampling pattern.

2. To investigate which validation methods can be used as a model selection tool to empirically assess the suitability of spatial proxies and select the most appropriate proxy configuration.

3. To provide guidance to practitioners regarding the use of spatial proxies in real-world applications.

We address the first two objectives in a simulation study, while for the third objective we carry out two case studies where we model air temperature and particulate air pollution in Spain. We further compare and discuss the findings in the context of the recently developed RF-GLS model to benchmark the performance of this alternative modelling approach.

## 2 Methods

### 2.1 Simulation study

We designed a simulation study on a virtual 300x100 grid to assess, in different prediction settings, the suitability of RF regression models using three different types of spatial proxies: coordinates, EDF, and RFsp (Fig. 1). Within the grid, two separate areas were defined (Fig. 1.1): sampling, from where observations were sampled and which coincided with the interpolation prediction area; and the extrapolation prediction area, used to evaluate spatial model transferability. The simulation consisted of the following steps:

1. We generated predictor and response surfaces (Fig. 1.1) according to the different scenarios described in Table 1: 1) "autocorrelated error", where residual autocorrelation is expected due to a spatially autocorrelated error term; 2) "complete", where no spatial autocorrelation is expected and therefore spatial proxies are assumed to be irrelevant; 3) "missing predictors", where residual autocorrelation is present due to missing predictors; and finally 4) "proxies only", where no predictors are available for modelling and only proxies are used. To generate the surfaces, unconditional sequential Gaussian simulation (Gebbers and de Bruin, 2010) was used to generate six independent predictor fields $X$ with 0 mean and
a spherical variogram with sill=1, nugget=0, and range equal to 10 or 40 (see examples in supporting Fig. A1) to be used in response $Y$ generation. Additionally, we simulated autocorrelated ($\mathcal{E}$, random field with 0 mean and a spherical variogram with sill=1, nugget=0, and range=25) and random ($\mathcal{E}'$, standard Gaussian) error surfaces (Fig. A1). We generated response surfaces using the equations in Table 1.

2. We simulated four sets of training points in the sampling area (Fig. 1.2) with a sample size of 200 following different distributions: regular samples were drawn by adding random noise (uniform distribution with parameters $U(-2, 2)$) to a regular grid, random samples were simulated via uniform random sampling, clustered samples were obtained by simulating 25 (weak clustering) or 10 (strong clustering) randomly-distributed parent points in a first step and 7 (weak) or 19 (strong) offspring points within a 8-unit (weak) or 6-unit (strong) buffer of each parent.

3. For each set of samples, we extracted the corresponding values of the response and predictors, deleted duplicate observations (i.e. two or more points intersecting with the same cell), and fitted a baseline RF model, which used predictors according to the corresponding scenario (Table 1). We also fitted coordinates, EDF, and RFsp models (see introduction for details) which included the predictors from the baseline model plus the spatial proxies (Fig. 1.3). We kept the number of trees at a constant value of 100 and tuned the hyperparameter `mtry` using out-of-bag samples and an equally-spaced
grid of length 5 ranging from 2 to the maximum number of predictors.

4. We used each of the fitted models to compute predictions for the entire area and calculated the "true" Root Mean Square Error (RMSE) by comparing the simulated and predicted response surfaces in all the interpolation and extrapolation areas separately (Fig. 1.4). In the baseline model for the "proxy only" scenario where no predictors were available, the mean of the response in the training data was used as a constant prediction. The expected minimum possible RMSE for scenarios 2-4 was equal to 1 (standard deviation of the random error), whereas it was equal to 0 for scenario "autocorrelated error" as the error could potentially be explained by the proxies.

5. Since the true RMSE is unknown in real-world applications, we also estimated the RMSE using additional validation methods (Fig. 1.5). First, a probability sample of 100 random test points was drawn and used to estimate the RMSE in the interpolation and extrapolation areas separately. Moreover, a 5-fold random CV and 5-fold kNNDM CV were used to estimate the RMSE. Briefly, kNNDM is a prediction-oriented method that provides predictive conditions in terms of geographical distances during CV similar to those encountered when using a model to predict a defined area (Linnenbrink et al., 2023; Milà et al., 2022). kNNDM has been shown to provide a better estimate for map accuracy than random k-fold CV when used with clustered samples, while returning fold configurations equivalent to random k-fold CV for regularly and randomly-distributed samples. Estimation of RMSE was done globally to account for the different fold sizes in kNNDM (Linnenbrink et al., 2023), i.e. we stacked all predictions in the different folds and computed the RMSE from all samples simultaneously, rather than computing the RMSE within each fold and then averaging. As kNNDM is dependent on the prediction objective, two different kNNDM configurations were used to estimate RMSE in the interpolation and extrapolation areas (Fig. 1.5).

6. We computed two additional metrics to understand the feature extrapolation potential and the variable importance of spatial proxies (Fig. 1.6). We calculated the percentage of the study area subject to feature extrapolation as per the Area of Applicability (AOA) (Meyer and Pebesma, 2021) using all training samples. AOA is defined as the area with feature values similar to those of the training data, and is computed based on distances in the predictor space. Unlike feature extrapolation metrics based on variable range or convex hulls, AOA takes into account predictor sparsity within the predictor range and weights variables by their importance in the model. Regarding variable importance, we used the mean decrease impurity method (Breiman, 2002) to quantify the percentage of the total average impurity decrease attributable to spatial proxies.

We ran 100 iterations of each simulation configuration, i.e. we fitted a total of 100 iterations $\times$ 4 prediction scenarios $\times$ 2 autocorrelation ranges $\times$ 4 sample distributions $\times$ 4 model types = 12,800 models (without counting the CV fits). We analysed the results of the simulations by examining the distributions of 1) the true RMSE, 2) the percentage of the study area subject to feature extrapolation, 3) the percentage of variable importance attributable to spatial proxies, and 4) the estimated RMSE; by each combination of simulation parameters and model type.

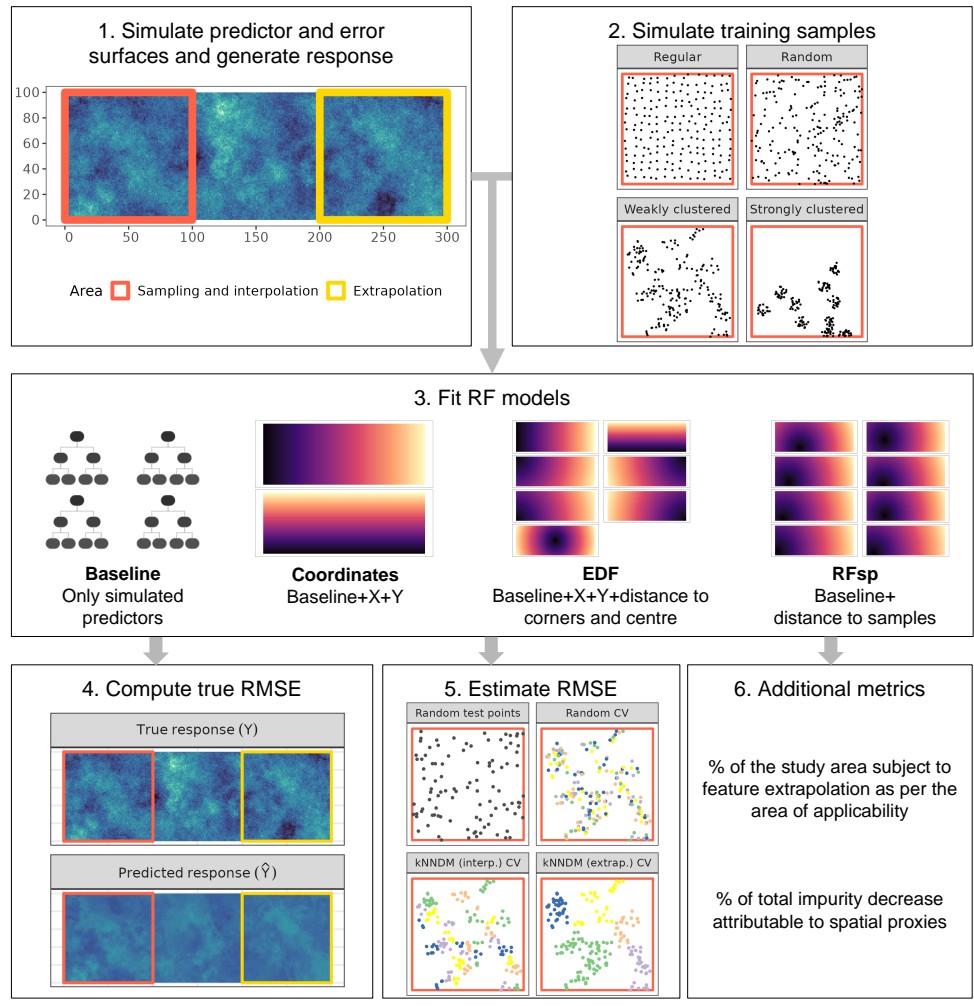

**Figure 1.** Workflow of the simulation study.

## 2.2 Comparison of spatial proxies with RF-GLS

As an alternative to spatial proxy approaches, we also tested the performance of the RF-GLS model recently proposed by Saha et al. (2023), an extension of RF which relaxes its independence assumption by accounting for spatial dependencies in several ways (see introduction for more details). To test the performance of RF-GLS, we included it in the set of candidate models, together with baseline and the three spatial proxy models, in the simulations presented in section 2.1, used it to predict the entire area, and computed the "true" RMSE in the interpolation and extrapolation areas by comparing the simulated and predicted response surfaces.

| Scenario | Description | Response generation equation | Predictors available for modelling |
|---|---|---|---|
| Autocorrelated error | All predictors are available, autocorrelated error | $Y = X_1 + X_2 \cdot X_3 + X_4 + X_5 \cdot X_6 + \mathcal{E}$ | $X_1, X_2, X_3, X_4, X_5, X_6$ |
| Complete | All predictors are available, random error | $Y = X_1 + X_2 \cdot X_3 + X_4 + X_5 \cdot X_6 + \mathcal{E}'$ | $X_1, X_2, X_3, X_4, X_5, X_6$ |
| Missing predictors | A subset of predictors are available, random error | $Y = X_1 + X_2 \cdot X_3 + X_4 + X_5 \cdot X_6 + \mathcal{E}'$ | $X_1, X_2, X_3$ |
| Proxies only | No predictors are available, random error | $Y = X_1 + X_2 \cdot X_3 + X_4 + X_5 \cdot X_6 + \mathcal{E}'$ | None |

**Table 1.** Description of the scenarios of the simulation study. $\mathcal{E}$ corresponds to a spatially autocorrelated error while $\mathcal{E}'$ is a random error.

## 2.3 Case studies

We modelled annual average air temperature and fine particulate air pollution for continental Spain in 2019 to examine the use of RF models with spatial proxies in real-word examples. For the first case study, we collected daily average air temperature data using the API of the *Agencia Española de Meteorología*, calculated station-based annual averages, and retained 195 stations with a temporal coverage of 75% or higher (Fig. 2). For the second, we collected data on concentrations of Particulate Matter with a diameter of 2.5 microns or less (PM$_{2.5}$) from the *Ministerio para la transición ecológica*. For PM$_{2.5}$ stations with hourly resolution, we first computed daily averages whenever at least 75% of the observations for a given day were available. Then, we computed annual averages and retained 124 stations with an annual temporal coverage of 75% or higher (Fig. 2).

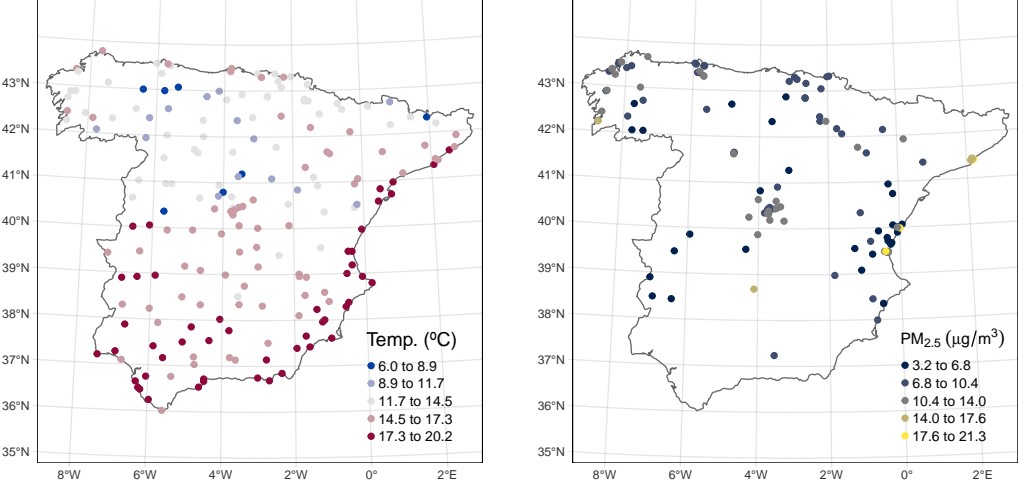

**Figure 2.** Spatial distribution of the reference station data for the air temperature and air pollution case studies.

We generated a 1 km × 1 km grid covering continental Spain as prediction area. Details of all data used for predictor generation are included in Table A1; while code for all pre-processing steps and processed data used for modelling are publicly

available (see code and data availability section below). Briefly, we collected a Digital Elevation Model (DEM), an impervious density product, gridded population counts, land cover data, coastline geometries, road geometries by type, a satellite-based Normalized Difference Vegetation Index (NDVI) from the MODIS Aqua 16-day NDVI product (MYD13A1) and 8-day Land Surface Temperature (LST, MYD11A2) products, annual NightTime Lights (NTL) from VIIRS, and European atmospheric composition reanalyses for $PM_{2.5}$ from Copernicus Atmosphere Monitoring Service (CAMS). We derived population density from the georeferenced population data; we computed % of different land cover classes (urban, industrial, agricultural, natural) in each 1km grid cell; we measured distances from each cell centroid to the nearest coastline; we calculated primary (highway and primary roads) and secondary (all other vehicle roads) road density as the length of the road segments within each 1 km $\times$ 1 km cell; we computed annual average composites of the NDVI, LST, and CAMS data. We regridded predictors to the target 1 km $\times$ 1 km grid using bilinear interpolation (downscaling) or averaging (upscaling) depending on the source resolution. We extracted predictor values at the station locations for subsequent modelling.

Unlike the simulation study, in these real-world case studies the strength of the spatial autocorrelation of the response and the sample spatial distribution were unknown. To understand how these factors may affect the performance of the different models, we performed an exploratory analysis for each response. First, we assessed the spatial distribution of the monitoring stations using exploratory spatial point pattern analyses. Namely, we estimated the empirical $\hat{G}$, $\hat{F}$, and $\hat{K}$ functions; Monte Carlo simulation (n=99) was used to construct simultaneous envelopes to assess departure from complete spatial randomness (Baddeley et al., 2015). Secondly, we computed empirical variograms of the response variables to assess the strength of the autocorrelation.

For each response, we considered two different sets of variables to be included in the models. First, a naive model, where only one predictor, known a priori to be a strong driver of the response, was used: elevation for temperature and primary road density for $PM_{2.5}$. Second, a complete model, where a much more comprehensive set of predictors was used (see list in supporting Table A1). Our motivation for the naive model was to examine whether spatial proxies could help explaining residual spatial autocorrelation due to missing predictors and therefore be used in predictor scarcity settings. Similarly to the simulation study, we used a RF regression baseline model with the selected predictors, as well as coordinates, EDF, and RFsp as additional proxy predictors. We fixed the number of trees to 300 and tuned the parameter `mtry` using out-of-bag samples and an equally-spaced grid of length 10 ranging from 1 to the maximum number of predictors. Using the same methods as in the simulation study, we estimated the performance by estimating the RMSE and $R^2$ using 10-fold random and kNNDM CV (no probability test samples were available), calculated the percentage of the study area subject to extrapolation, and estimated the relative importance of spatial proxies. We plotted the predicted surfaces and presented the computed statistics. We assessed residual spatial autocorrelation using empirical variograms of the residuals of each model to evaluate whether spatial dependencies in the data had been captured.

## 2.4 Implementation

Our analyses were carried out in R version 4.2.2 (R Core Team, 2022) using several packages: `sf` (Pebesma, 2018) and `terra` (Hijmans, 2022) for spatial data management; `caret` (Kuhn, 2022), `ranger` (Wright and Ziegler, 2017), `RandomForestsGLS`

(Saha et al., 2022), and `CAST` (Meyer et al., 2023) for spatial modelling; `gstat` (Pebesma, 2004) for random field simulation; and `ggplot2` (Wickham, 2016) and `tmap` (Tennekes, 2018) for graphics and cartographic representations. Additional packages were used for other minor tasks.

## 3 Results

### 3.1 Simulation study

#### 3.1.1 Suitability of spatial proxies

The prediction objective was a clear determinant of the suitability of spatial proxies. When aiming to predict in the extrapolation area (Fig. 3), baseline models always outperformed spatial proxy models regardless of the other parameters, highlighting the lack of ability of proxies to successfully transfer to new areas different to those where they were trained. This was supported by feature extrapolation statistics of proxy models (supporting Fig. A2), which indicated that a very large part or even all of the extrapolation area had feature values not covered by the training data.

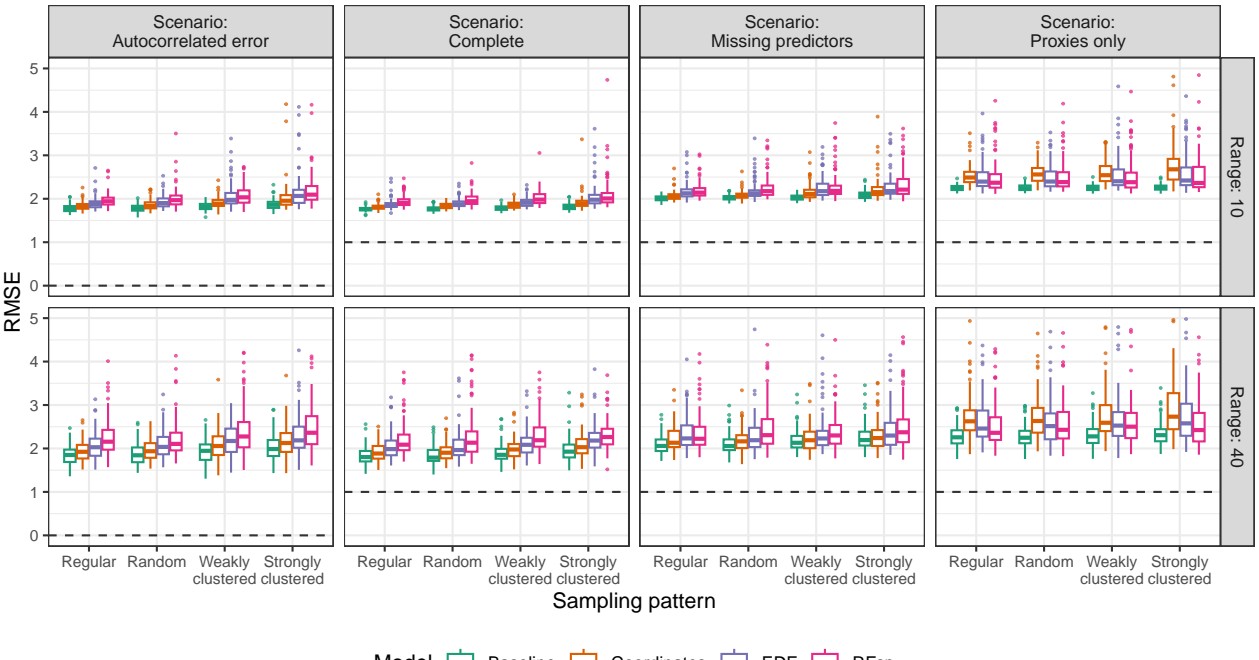

**Figure 3.** True RMSE in the extrapolation area of each model type by scenario, autocorrelation range, and sampling pattern. The dashed line indicates the minimum possible RMSE for each scenario. RMSE for the baseline model in the "proxies only" scenario uses a constant prediction calculated as the average response value in the training data. Outliers larger than 5 are not shown for visualization purposes.

The suitability of spatial proxies for interpolation was more complex and depended on a series of additional factors, including the strength of residual autocorrelation (Fig. 4). In the "complete" scenario where residual spatial autocorrelation was not expected, models with spatial proxies yielded RMSE values that were similar or larger than the respective baseline models. On the other hand, in scenarios where residual autocorrelation was expected either due to an autocorrelated error term or missing predictors, models with spatial proxies showed smaller errors in many instances. Regarding the extent of the spatial autocorrelation, spatial proxy models offered more benefits in situations in which the spatial structure of the predictors and response, expressed as the autocorrelation range, was stronger.

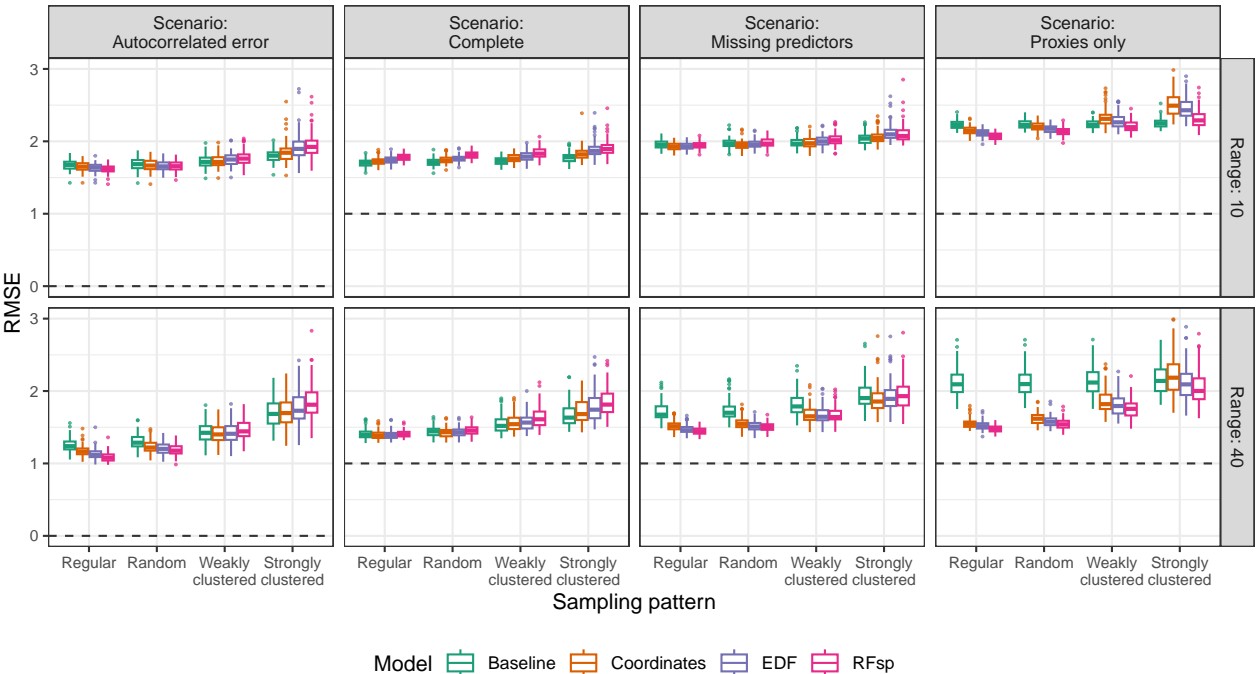

**Figure 4.** True RMSE in the interpolation area of each model type by scenario, autocorrelation range, and sampling pattern. The dashed line indicates the minimum possible RMSE for each scenario. RMSE for the baseline model in the "proxies only" scenario uses a constant prediction calculated as the average response value in the training data. Outliers larger than 3 are not shown for visualization purposes.

The suitability of spatial proxies for interpolation was also influenced by the sampling pattern. With random and regular samples (Fig. 4), the addition of spatial proxies tended to decrease errors in scenarios where residual spatial autocorrelation was expected, while yielding comparable or only slightly worse results in the "complete" scenario. This is connected to the low feature extrapolation observed for random and regular sampling patterns (supporting Fig. A3): since samples covered the whole extent of the interpolation area, adding spatial proxies did not impact feature extrapolation, which remained low. Nonetheless, when samples were clustered, the addition of spatial proxies increased feature extrapolation (supporting Fig. A3) leading to models with a generally larger RMSE compared to baseline models, except in cases where the residual spatial autocorrelation was strong and the sampling pattern was only weakly clustered (see "missing predictors" scenario with weakly

clustered samples and range 40 in Fig. 4). Finally, interpolation models using only spatial proxies as predictors performed

nearly as well as models with all (scenario: complete) or a subset (scenario: missing predictors) of predictors provided samples were regularly or randomly distributed and the autocorrelation range was 40 (Fig. 4).

Comparing the different types of spatial proxies, whenever their use was not appropriate for either interpolation or extrapolation, RFsp tended to give worse results than coordinates; nonetheless, together with EDF, it also yielded the largest gains when the use of proxies was beneficial. We attribute this to the larger number of spatial proxy predictors in RFsp and EDF

models compared to coordinates, leading to a larger proxy feature importance (supporting Fig. A4). Feature importance of spatial proxies was larger for clustered samples compared to regular and random patterns, as well as for the long autocorrelation range (supporting Fig. A4).

### 3.1.2 Validation methods for proxy selection

In the extrapolation area and for the "autocorrelated error" scenario, random 5-fold CV did not only severely underestimate the

true RMSE, but also systematically and erroneously suggested that models with proxies had a similar or superior performance compared to baseline models (Fig. 5). On the other hand, both probability test samples and kNNDM CV correctly ranked models according to their true RMSE. Results in the extrapolation area for the rest of scenarios are available in supporting Figs. A5-A7 and showed similar patterns.

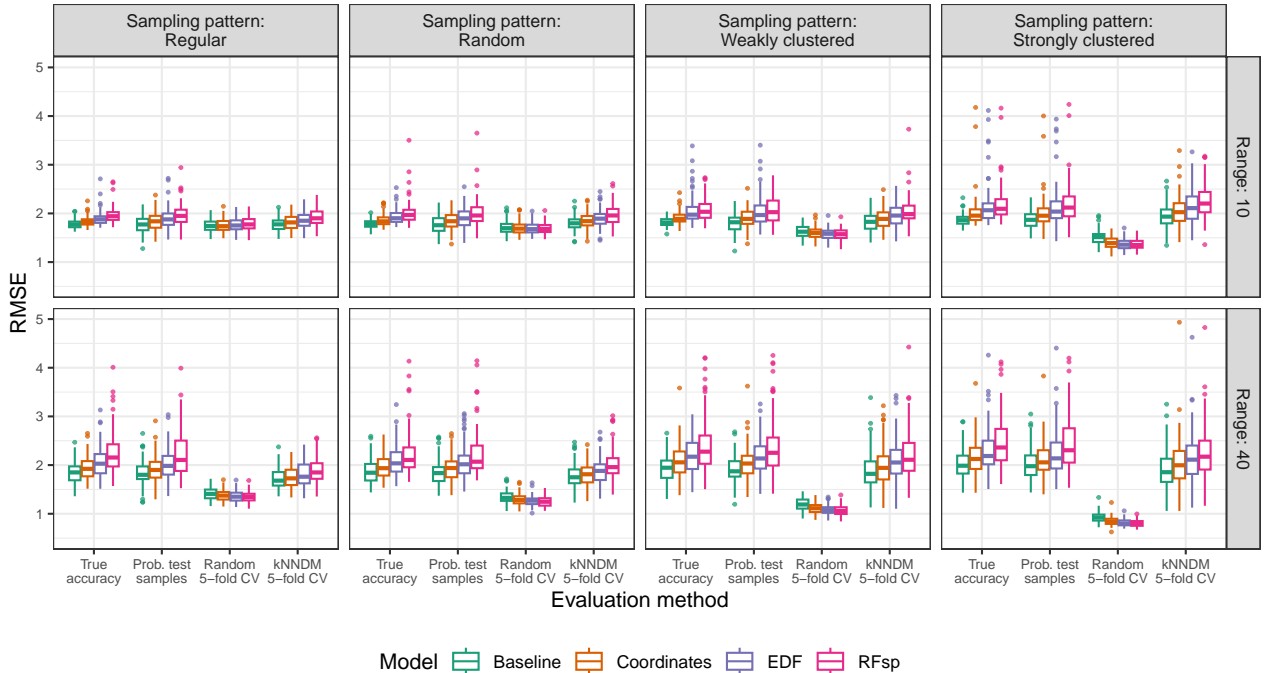

**Figure 5.** True and estimated RMSE in the extrapolation area and the "autocorrelated error" scenario by evaluation method, autocorrelation range, and sampling pattern. Outliers larger than 5 are not shown for visualization purposes.

In the interpolation area and the "autocorrelated error" scenario (Fig. 6), all validation methods correctly ranked models
under regular and random sampling patterns. However, under clustered sampling patterns, random k-fold CV indicated that
models with spatial proxies were superior when in fact they were similar or worse. Similar results were observed for the rest
of scenarios in the interpolation area (supporting Figs. A8-A10).

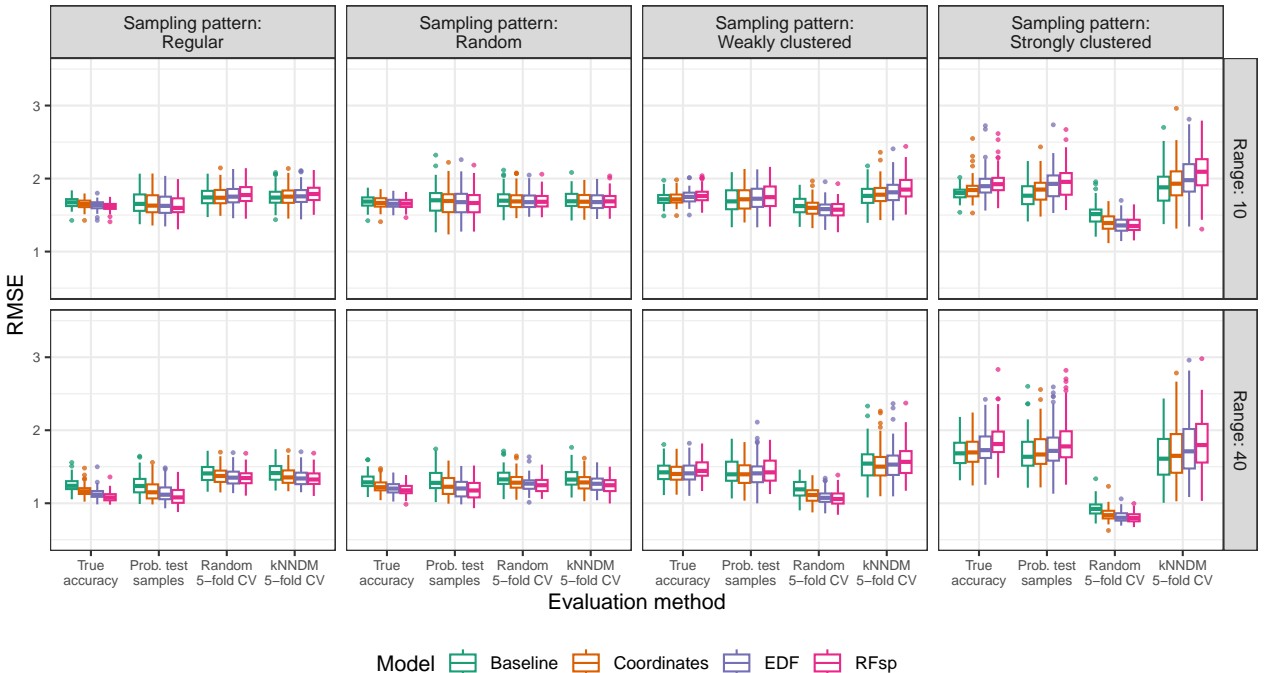

**Figure 6.** True and estimated RMSE in the interpolation area and the "autocorrelated error" scenario by evaluation method, autocorrelation
range, and sampling pattern. Outliers larger than 3.5 are not shown for visualization purposes.

### 3.1.3 Comparison of spatial proxies with RF-GLS

RF-GLS outperformed or was on a par with the best-performing standard RF model with and without proxies for all parameter
combinations in both the interpolation (Fig. 7) and extrapolation (supporting Fig. A11) areas in the simulation study. The most
relevant gains in performance when comparing RF-GLS to RF with and without proxies were in the "autocorrelated error"
scenario for the interpolation area with regular and random samples, for which RMSE were substantially lower.

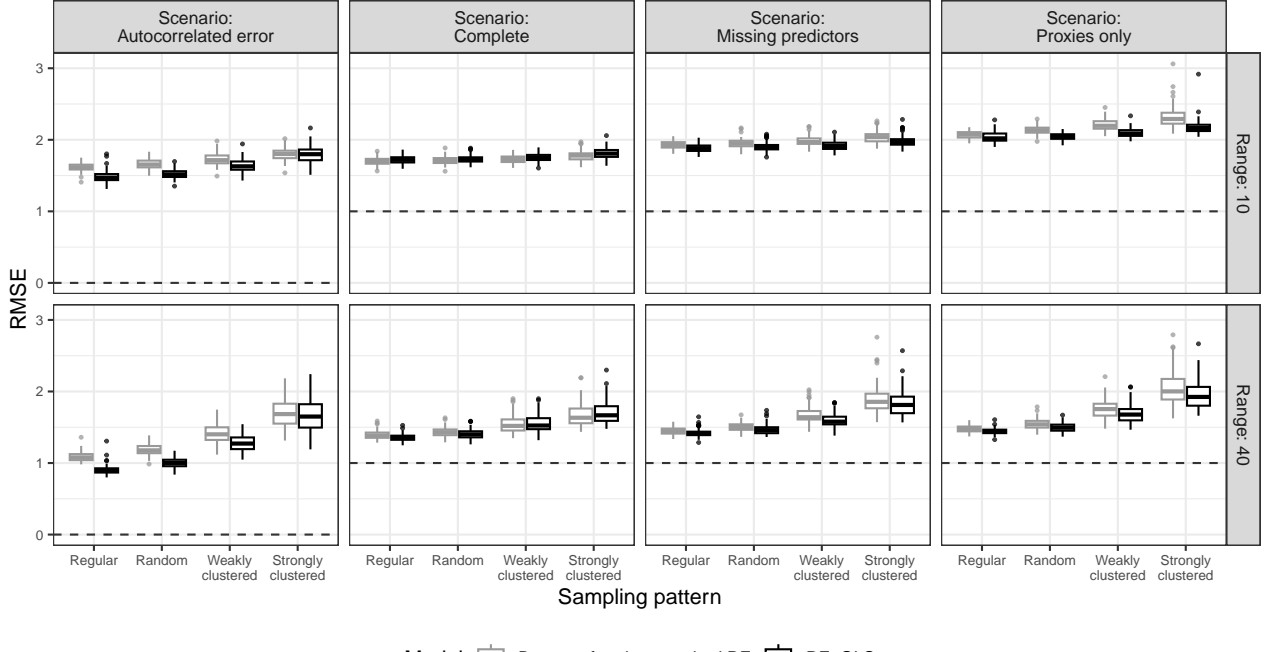

**Figure 7.** True RMSE in the interpolation area of the best-performing standard RF for each parameter combination (i.e. the standard RF model with/without proxies with the lowest median RMSE) and RF-GLS, by prediction scenario, spatial autocorrelation range, and sampling pattern. The dashed line indicates the minimum possible RMSE for each scenario.

## 3.2 Case studies

Air temperature meteorological stations were well spread over the study area (Fig. 2) and the point pattern exploratory analysis
did not suggest a major departure from complete spatial randomness, although there was some evidence of a regular pattern (supporting Fig. A12). Aligned with these results, kNNDM generalised to a random 10-fold CV (supporting Fig. A13).

Results for the naive temperature model indicated substantial gains in performance when using spatial proxies, which yielded only slightly worse results than complete models (Table 2). Performance of all complete models was similar. Feature extrapolation was similar in all cases and smaller than 10% of the study area. We detected strong spatial autocorrelation in the response
and the residuals of the naive baseline model, which mostly disappeared when adding the whole set of predictors and/or spatial proxies (supporting Fig. A14). Adding spatial proxies to the baseline naive model with only a DEM resulted in different patterns and smoother predicted surfaces (Fig. 8). Comparing naive models with spatial proxies and complete models, spatial patterns were quite similar but more local variation could be appreciated in the latter. Differences between maps derived from complete models with and without proxies were minor.

| Model | RMSE$_{random}$ (ºC) | R$^2_{random}$ | RMSE$_{kNNDM}$ (ºC) | R$^2_{kNNDM}$ | Extrapolation (%) | Proxy importance (%) |
|---|---|---|---|---|---|---|
| Naive | | | | | | |
| Baseline | 2.02 (0.27) | 0.49 (0.2) | 2.02 | 0.51 | 8.47 | 0.00 |
| Coordinates | 0.93 (0.29) | 0.88 (0.07) | 0.91 | 0.90 | 5.29 | 49.86 |
| EDF | 0.93 (0.29) | 0.89 (0.07) | 0.92 | 0.89 | 6.00 | 53.56 |
| RFsp | 1.03 (0.3) | 0.87 (0.07) | 1.01 | 0.87 | 6.40 | 63.33 |
| Complete | | | | | | |
| Baseline | 0.81 (0.21) | 0.92 (0.04) | 0.82 | 0.92 | 7.25 | 0.00 |
| Coordinates | 0.77 (0.28) | 0.93 (0.04) | 0.79 | 0.93 | 8.80 | 19.14 |
| EDF | 0.8 (0.27) | 0.92 (0.05) | 0.80 | 0.92 | 6.33 | 22.89 |
| RFsp | 0.85 (0.23) | 0.92 (0.04) | 0.86 | 0.91 | 6.91 | 29.65 |

**Table 2.** Results of the temperature case study. Subscripts for RMSE and R$^2$ indicate the type of 10-fold CV used to compute the statistics. Random 10-fold CV statistics are computed as the mean (SD) of the statistic calculated in each fold, while kNNDM CV statistics were computed by stacking all observed and predicted values (see methods).

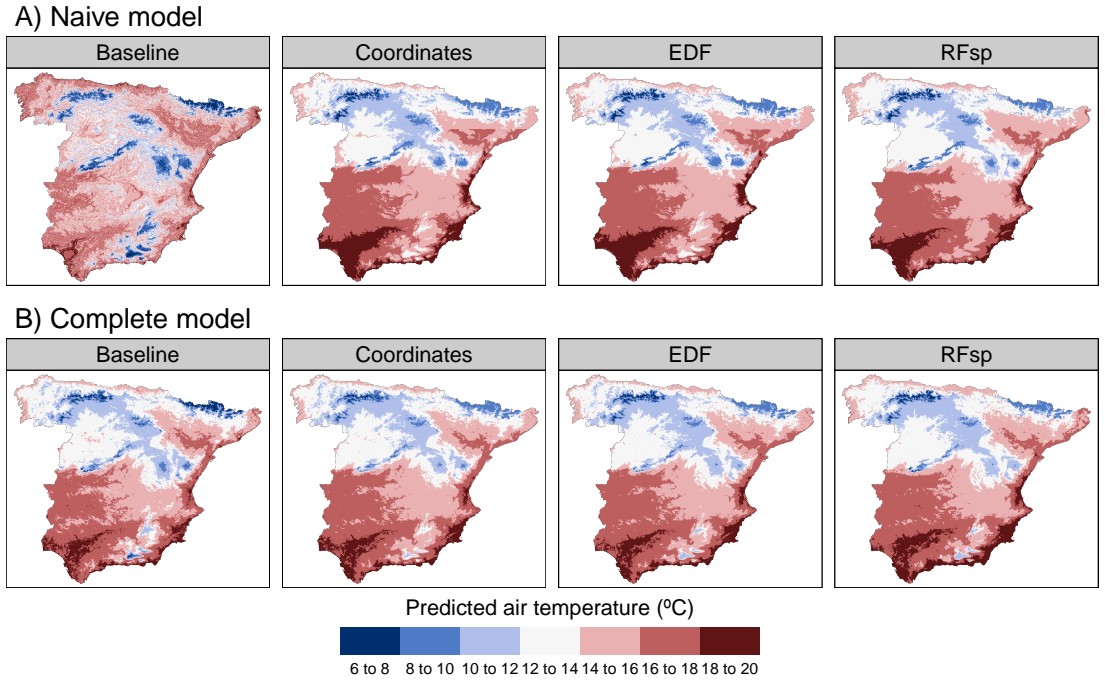

**Figure 8.** Predicted air temperature using A) naive (DEM only) and B) complete predictors by model type.

The distribution of PM$_{2.5}$ stations visually appeared to be spatially clustered (Fig. 2), which was confirmed by the exploratory spatial point pattern analysis with a clear departure from complete spatial randomness (supporting Fig. A15). Reflecting the clustering pattern, the resulting kNNDM had a distinct spatial configuration (supporting Fig. A16).

According to random 10-fold CV, the estimated performance of the baseline naive model in terms of $R^2$ was almost null, but improved substantially when adding spatial proxies. Nonetheless, when using kNNDM CV, the estimated performance was

similarly null in all cases (Table 3). Estimated RMSEs of complete models were still lower when using random vs. kNNDM CV; however, statistics across the different model types were much more similar. Feature extrapolation was the highest in naive models, where proxies had a larger importance that translated into mapping artefacts that were especially evident in the coordinates model (Fig. 9). Unlike the temperature case study, the predicted surfaces of naive models with proxies and complete models were very different, suggesting that the added geographical predictors could not successfully account for the

missing predictors. Prediction maps for complete models with different spatial proxies were much more similar. Inspection of the empirical variograms for the response and residuals of the naive baseline model indicated presence of spatial autocorrelation that was weaker than for air temperature, and which disappeared in complete and spatial proxy models (supporting Fig. A17).

| Model | $RMSE_{random}$ ($\mu g/m^3$) | $R^2_{random}$ | $RMSE_{kNNDM}$ ($\mu g/m^3$) | $R^2_{kNNDM}$ | Extrapolation (%) | Proxy importance (%) |
|---|---|---|---|---|---|---|
| Naive | | | | | | |
| Baseline | 3.6 (1.03) | 0.13 (0.18) | 3.76 | 0.02 | 1.54 | 0.00 |
| Coordinates | 2.69 (0.52) | 0.37 (0.26) | 3.60 | 0.04 | 13.52 | 78.85 |
| EDF | 2.6 (0.63) | 0.43 (0.27) | 3.65 | 0.04 | 17.42 | 90.11 |
| RFsp | 2.64 (0.75) | 0.44 (0.28) | 3.94 | 0.01 | 9.58 | 94.76 |
| Complete | | | | | | |
| Baseline | 2.5 (0.51) | 0.46 (0.22) | 3.00 | 0.30 | 0.65 | 0.00 |
| Coordinates | 2.41 (0.54) | 0.49 (0.23) | 2.99 | 0.31 | 7.03 | 22.88 |
| EDF | 2.43 (0.55) | 0.48 (0.24) | 3.04 | 0.29 | 9.41 | 36.16 |
| RFsp | 2.39 (0.59) | 0.49 (0.26) | 3.33 | 0.17 | 3.39 | 58.90 |

**Table 3.** Results of the PM$_{2.5}$ case study. Subscripts for RMSE and $R^2$ indicate the type of 10-fold CV used to compute the statistics. Random 10-fold CV statistics are computed as the mean (SD) of the statistic calculated in each fold, while kNNDM CV statistics were computed by stacking all observed and predicted values (see methods).

## 4 Discussion

Our first objective was to assess the suitability of spatial proxies depending on the modelling objective, the strength of the

residual spatial autocorrelation, and the sampling pattern. Regarding the modelling objective, we found that a RF with spatial proxies is never beneficial when the goal is spatial model transfer to a new area. By adding spatial proxies to the predictor set that identify specific locations of the sampling area, we inevitably face feature extrapolation in the new area as values of proxy predictors will be completely different. Not only that, but when proxies are used as node-splitting variables in the RF, we end up only using observations placed on the edge of the sampling area regardless of the distance to the new prediction area,

unlike methods such as RF-GLS or regression kriging that can account for the autocorrelation decay with increasing distances.

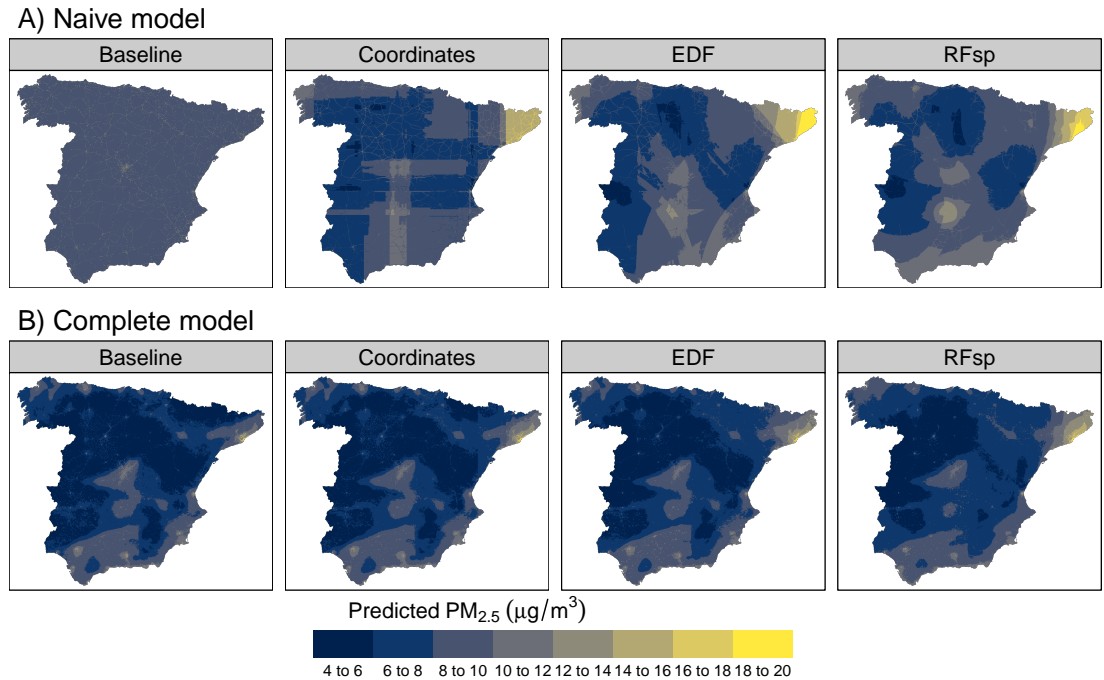

**Figure 9.** Predicted PM$_{2.5}$ using A) naive (primary road density only) and B) complete predictors by model type.

Therefore, these variables should not be used for prediction in new geographical areas and the focus should be placed on causal predictors.

For interpolation purposes, however, proxies may be beneficial depending on additional factors. We discovered that one of the conditions that make the inclusion of spatial proxies in RF models to be beneficial is the presence of residual autocorrelation

due to missing predictors or an autocorrelated error. These potential benefits can be understood by the capacity of spatial proxies to account for residual spatial autocorrelation (Hengl et al., 2018; Behrens et al., 2018), which our results confirmed both in terms of improved performance and removed residual autocorrelation, especially when using a larger number of proxies (EDF or RFsp). However, in complete models with no residual autocorrelation, the similar or sometimes worse performance is due to adding an irrelevant set of predictors that are noise to the model. Unlike regression kriging, where spatial autocorrelation is

modelled in the residuals and in its absence would result in a pure nugget effect, i.e. a flat variogram leading to an ordinary least squares estimation (Hengl, 2007), in a ML model the irrelevant proxies are still included. Even though RF is fairly robust to the addition of irrelevant predictors (Kuhn and Johnson, 2019), a decrease in performance was sometimes observed. In addition to the presence of spatial autocorrelation, the strength of the spatial structure as defined by the autocorrelation range was also important. When ranges become shorter, we get closer to the independence assumption of a non-spatial model and thus proxies

start to become irrelevant. Experiments for response variables with weaker spatial autocorrelation such as land cover would be interesting follow-up studies to further clarify this point.

In addition to the presence of significant spatial autocorrelation, we found that an almost necessary condition for proxies to be beneficial for interpolation is to have regular or randomly-distributed samples. This is not surprising since the feature extrapolation potential of spatial proxies with clustered samples has been stressed before (Meyer et al., 2019; Hengl et al., 2018; Cracknell and Reading, 2014). The more proxies used in the models, the larger the feature extrapolation was. Given these results, although it would be required that spatial proxies had a lower importance when used with clustered samples vs. regular or random, we actually observed the opposite. This is likely a sign of overfitting, where the model uses the proxies to determine the position of the sampling clusters (Meyer et al., 2019), a hypothesis that the difference between the estimated random CV, and probability test samples and kNNDM CV, supported. Our results are consistent with spatial sampling recommendations for ML models such as RF, which suggest using designs that ensure a good spread in the most important predictors to optimise performance (Wadoux et al., 2019). Hence, spatial proxies are expected to be poorly suited for modelling with clustered samples by design. Even though our simulations indicate that weakly clustered data may sometimes also slightly benefit from spatial proxies, we recommend to proceed with caution because it is challenging to define the degree of clustering at which they start to be harmful.

Our simulations allow us to give general guidelines on the adequacy of spatial proxies; however, it is important to have a way to confirm them empirically. This was the focus of the second objective, for which we showed that random CV underestimates map accuracy when assessing extrapolation performance or interpolation with clustered samples, which has been shown before (Linnenbrink et al., 2023; Wadoux et al., 2021). Perhaps even more important, random CV incorrectly ranks models in those instances, systematically favouring models with proxies even though those are not always appropriate. On the other hand, probability test samples and kNNDM CV did provide correct model ranks. We think this is related to overfitting and the inability of random k-fold CV to reflect predictive conditions (Meyer and Pebesma, 2022): in the presence of clustered sampling, adding spatial proxies may actually help the model to predict at locations geographically close to the samples as reflected by random CV, yet fail to generalise to the entire prediction area as measured by probability test samples and kNNDM.

Our additional analyses regarding the RF-GLS model proposed by Saha et al. (2023) indicate that it performed equally or better than the best-performing standard RF with/without spatial proxies in all parameter configurations, which we attribute to several reasons. First, in RF-GLS residual variability is modelled as a Gaussian process rather than with spatial proxy predictors in the mean term, which minimizes feature extrapolation and spatial overfitting problems in spatial model transfer or interpolation with clustered samples. Furthermore, in RF-GLS the independence assumption of RF is relaxed as spatial autocorrelation is accounted for during the model fitting. Finally, RF-GLS can adapt better to settings where residual spatial autocorrelation is weak or absent since the estimation of the covariance function can take the absence of autocorrelation into account. Hence, we think that RF-GLS is a step forward in creating truly spatial ML models, and it should be considered as a candidate algorithm for spatial prediction tasks.

As the third objective, we presented two case studies with distinct characteristics that reflect different real-world settings. For air temperature, stations were spread across all the prediction area and measurements exhibited strong spatial autocorrelation. We found that a model with only a DEM and spatial proxies managed to account for the residual spatial autocorrelation, and performed almost as well as a much more comprehensive model which produced similar predicted surfaces. This highlights

the value of spatial proxies for cost-effective predictive modelling as long as the conditions outlined above are met. Regarding air pollution, samples were clustered and the autocorrelation was weaker. In both naive and complete models, spatial proxies did not improve the performance and large differences in the CV approaches were revealed, highlighting the aforementioned risk of spatial overfitting and wrong conclusions when inappropriate validation practices are used. In the two case studies, we showed the importance of performing a comprehensive spatial exploratory analysis to determine the sample distribution and the response and residual spatial autocorrelation in the baseline model (i.e. without proxies). The results of this analysis can help us determine whether a spatial proxy approach is advisable a priori, which can be confirmed a posteriori using model selection tools such as probability test samples or kNNDM CV.

In this study, we included a wide range of conditions typically encountered in environmental spatial modelling. Nonetheless, there are several points for future work. First, we focused on RF regression and, while we think that our results likely extend to other ML algorithms, the extrapolation behaviour and sensitivity to irrelevant predictors differs by algorithm and might limit the ability to generalize our results. Second, our analysis was based on the adequacy of spatial proxies from a prediction accuracy point of view. When using RF for knowledge discovery, variables with long or infinite autocorrelation ranges such as spatial proxies have been identified to be beyond the prediction horizon (Behrens and Viscarra Rossel, 2020; Wadoux et al., 2020b; Fourcade et al., 2018) and variable importance statistics in models including them should be interpreted with extreme caution (Meyer et al., 2019; Wadoux et al., 2020a). Third, feature selection based on an appropriate CV scheme has been shown to be helpful to discard irrelevant features prone to overfitting that generalise poorly to new locations such as coordinates (Meyer et al., 2019). In future work, it would be interesting to explore whether feature selection could help to identify irrelevant spatial proxy features. Fourth, we focused our investigation on the potential of spatial proxies to account for spatial autocorrelation while it has been suggested that coordinate and distance fields can also be useful to account for non-stationarity (Behrens and Viscarra Rossel, 2020), which remains to be explored. Finally, the scope of our study was limited to spatial proxies approaches and RF-GLS; however, our analyses could be extended to other models proposed in the literature, e.g. models including spatial lags of the response as prediction features (Sekulić et al., 2020).

## 5 Conclusions

We recommend RF with spatial proxies in cases where all of these conditions apply: 1) the sampling and prediction areas overlap (i.e. spatial interpolation), 2) there is presence of significant residual spatial autocorrelation due to missing predictors or an autocorrelated error term, and 3) samples are regularly or randomly distributed over the prediction area. In such cases, the addition of spatial proxies is very likely to be beneficial in terms or performance. If samples are regular or randomly-distributed but no residual autocorrelation is present, the addition of spatial proxies will have little impact. Finally, in the presence of clustered samples, using spatial proxies in RF models is generally not recommended since their inclusion can degrade model performance especially if residual autocorrelation is weak and the clustering is strong. Proxies should not be used for spatial model transfer.

More generally, we have shown that the benefits of RF with spatial proxies are not universal and therefore it should not be taken as a default approach without careful consideration. Spatial exploratory analysis of the sample distribution and the response and residual autocorrelation are recommended as preliminary steps to evaluate the suitability of spatial proxies, while probability test samples and kNNDM CV can be used as model selection tools to confirm it and choose the best set of proxies. Random k-fold CV should not be used for model selection if the objective is spatial model transfer or in the presence of clustered samples, since it erroneously favours models with spatial proxies. RF-GLS should be considered as a candidate modelling algorithm for spatial prediction tasks.

*Code and data availability.* The code for the analysis and the presentation of the results, as well as the data used in the case studies, are available at Milà (2024).

## Appendix A: Supplementary figures and tables

**Figure A1.** Example realizations of random fields used in the simulation study. All random fields have a 0 mean; predictor and autocorrelated error surfaces were generated using unconditional simulation with a spherical variogram with sill=1, nugget=0, and range indicated in the panel; random error was generated using a standard Gaussian distribution without spatial autocorrelation.

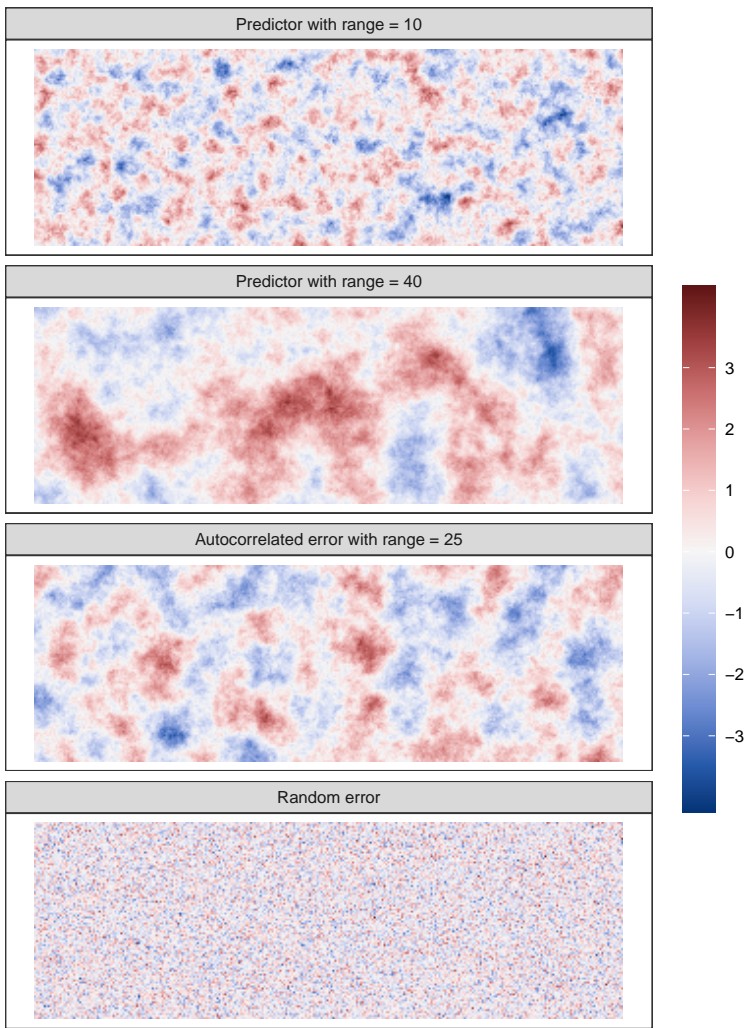

**Figure A2.** Feature extrapolation expressed as the percentage of the extrapolation prediction area outside of the Area of Applicability (AOA) of each model type by prediction scenario, spatial autocorrelation range, and sampling pattern.

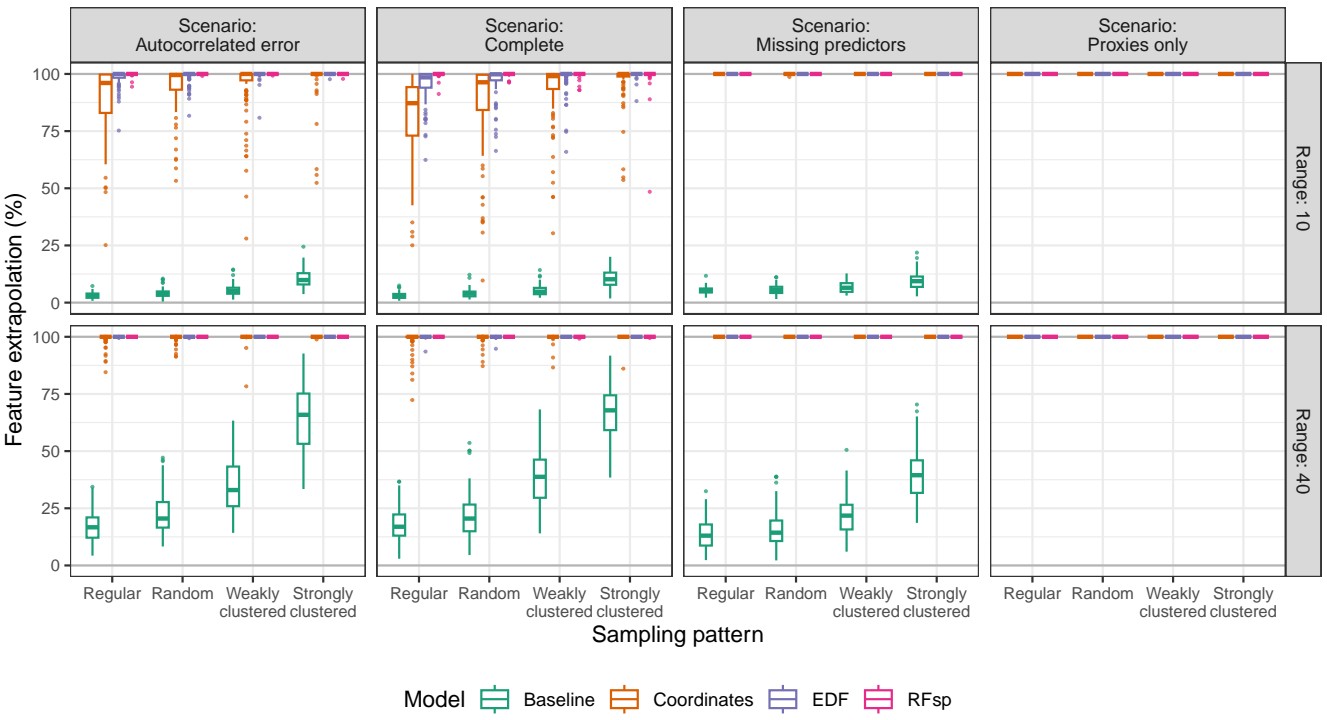

**Figure A3.** Feature extrapolation expressed as the percentage of the interpolation prediction area outside of the Area of Applicability (AOA) of each model type by prediction scenario, spatial autocorrelation range, and sampling pattern.

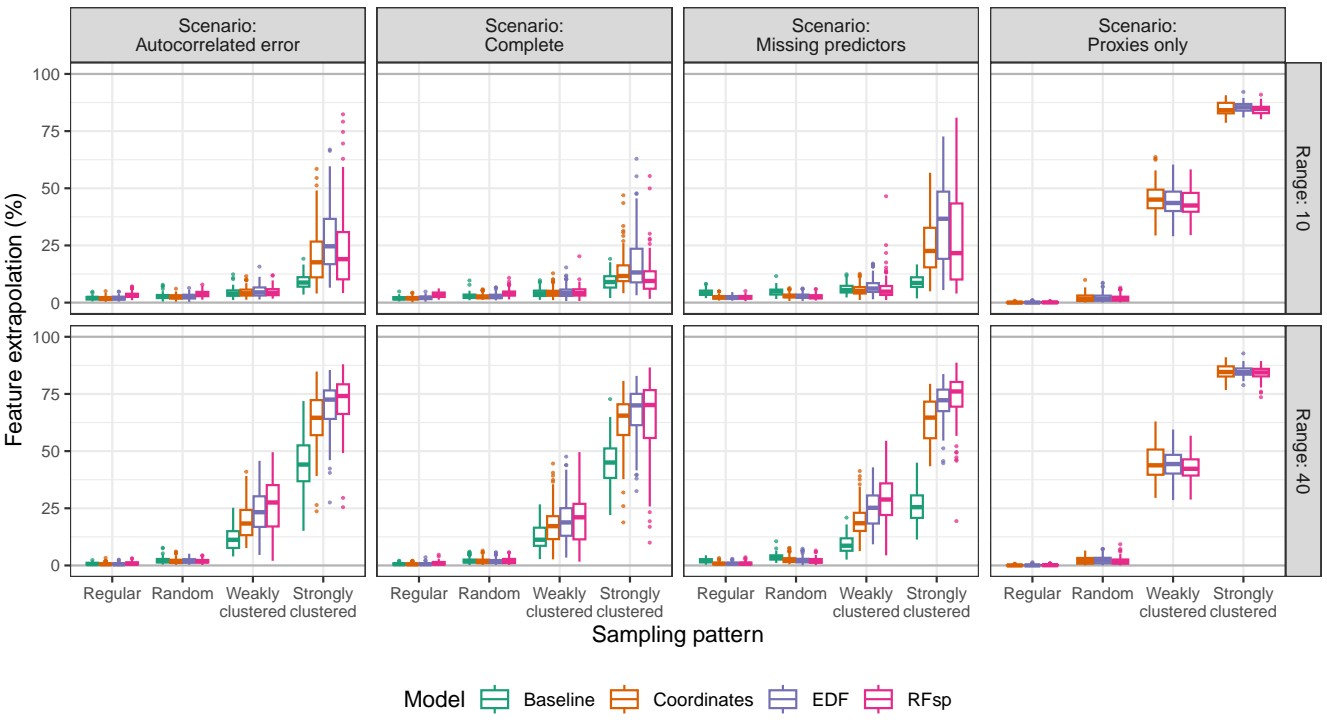

**Figure A4.** Variable importance of spatial proxies expressed as the percentage of total mean impurity decrease attributable to those variables for each model type by prediction scenario, spatial autocorrelation range, and sampling pattern.

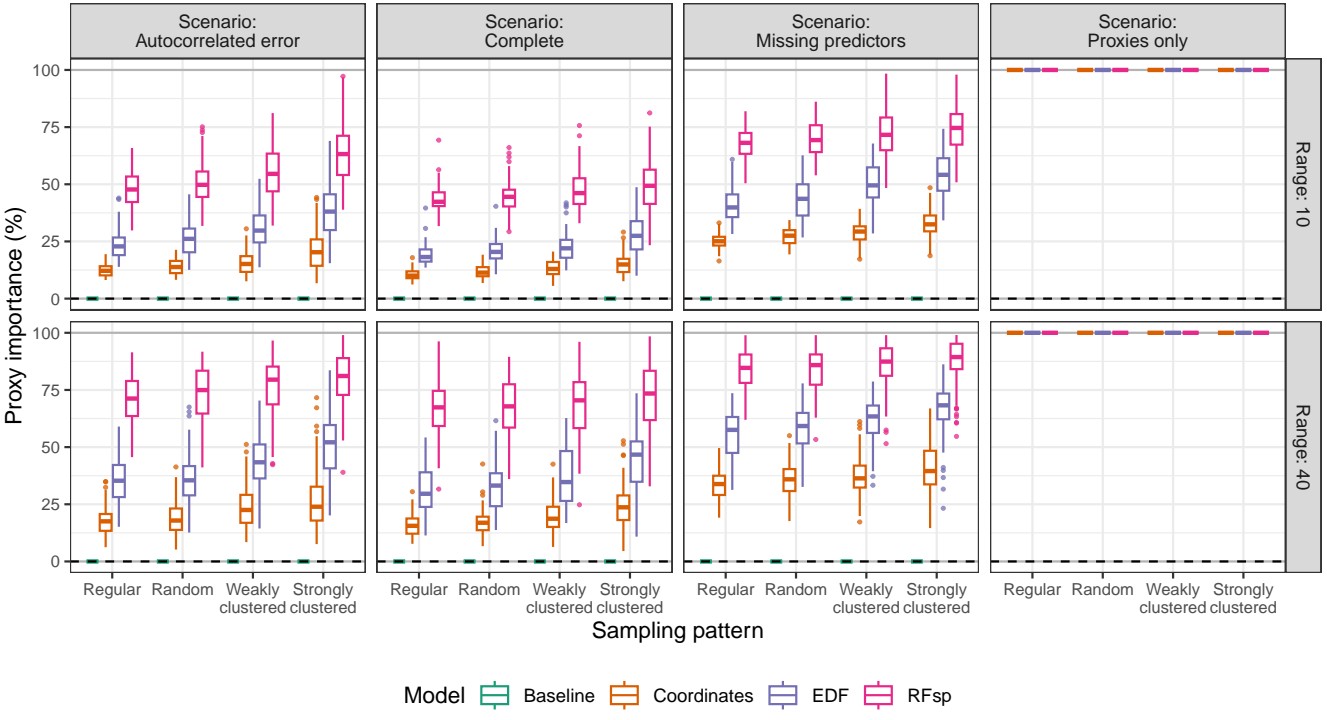

**Figure A5.** True and estimated RMSE in the extrapolation area and the "complete" scenario by evaluation method, autocorrelation range, and sampling pattern. Outliers larger than 4 are not shown for visualization purposes.

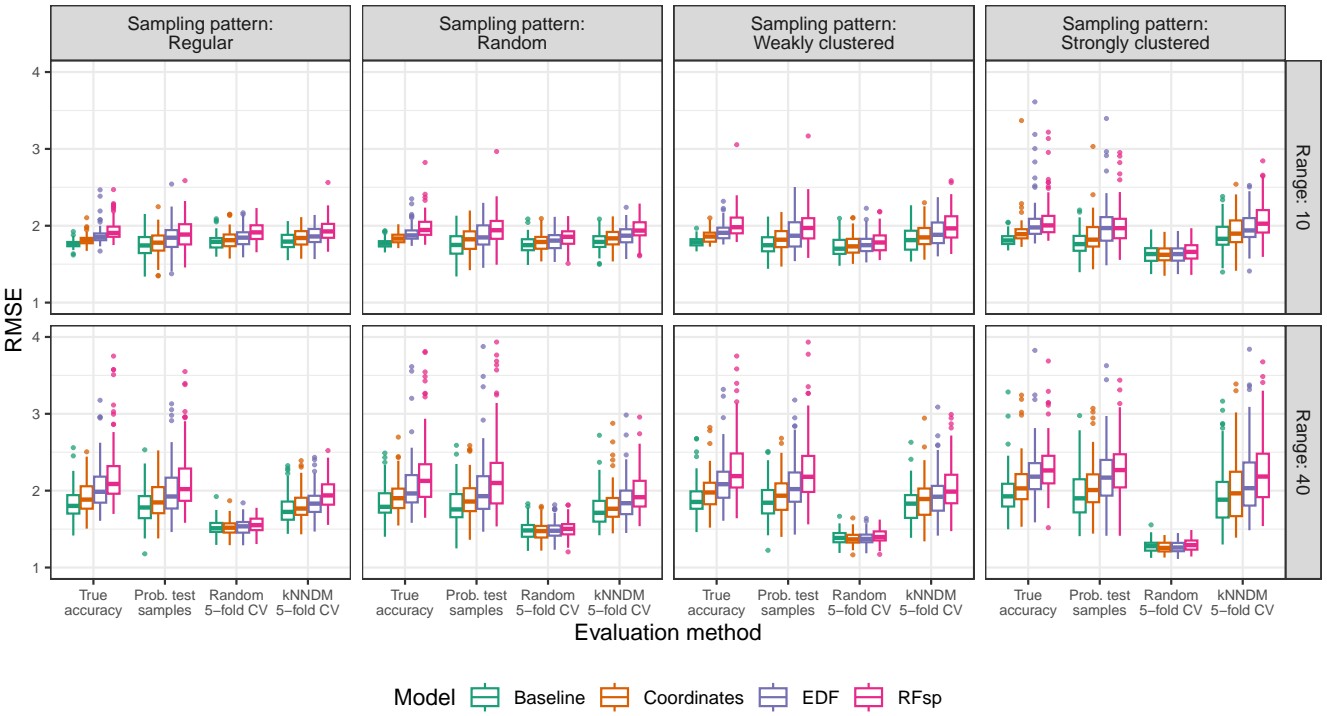

**Figure A6.** True and estimated RMSE in the extrapolation area and the "missing predictors" scenario by evaluation method, autocorrelation range, and sampling pattern. Outliers larger than 5 are not shown for visualization purposes.

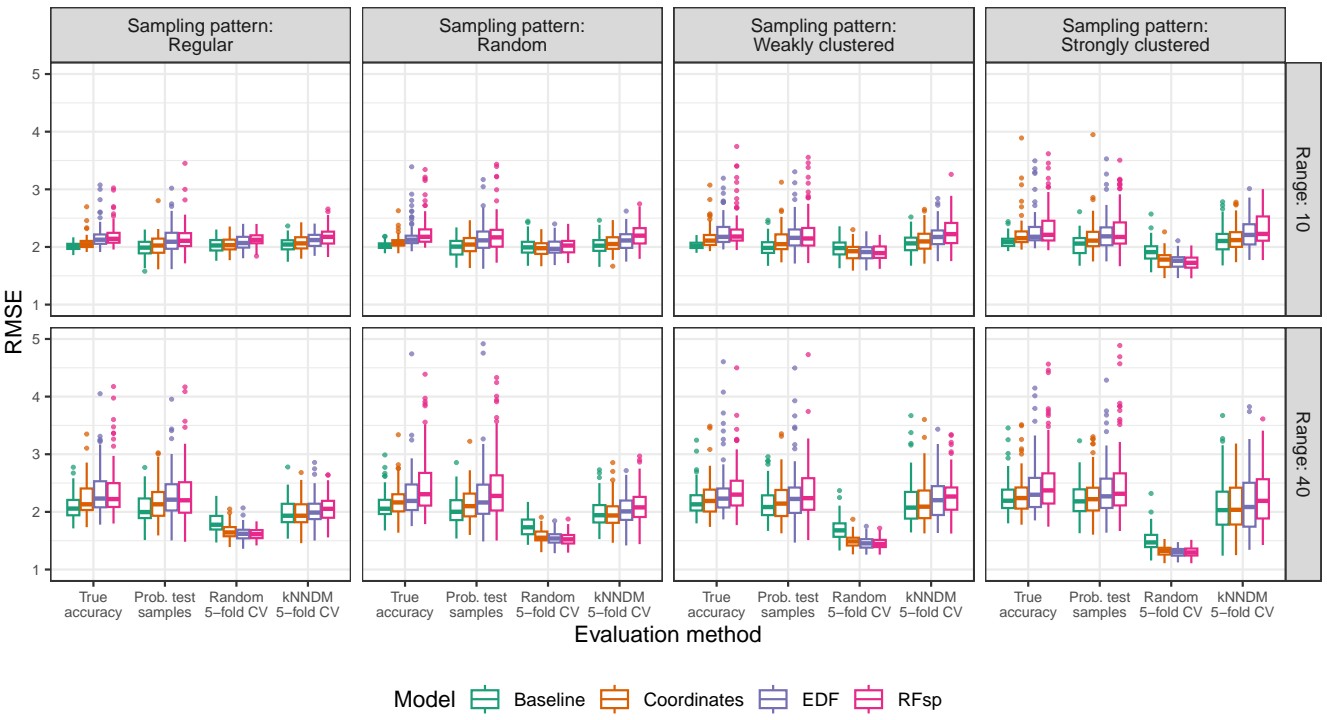

**Figure A7.** True and estimated RMSE in the extrapolation area and the "proxies only" scenario by evaluation method, autocorrelation range, and sampling pattern. Results for the baseline model were not calculated as no predictors were available for modelling. Outliers larger than 6 are not shown for visualization purposes.

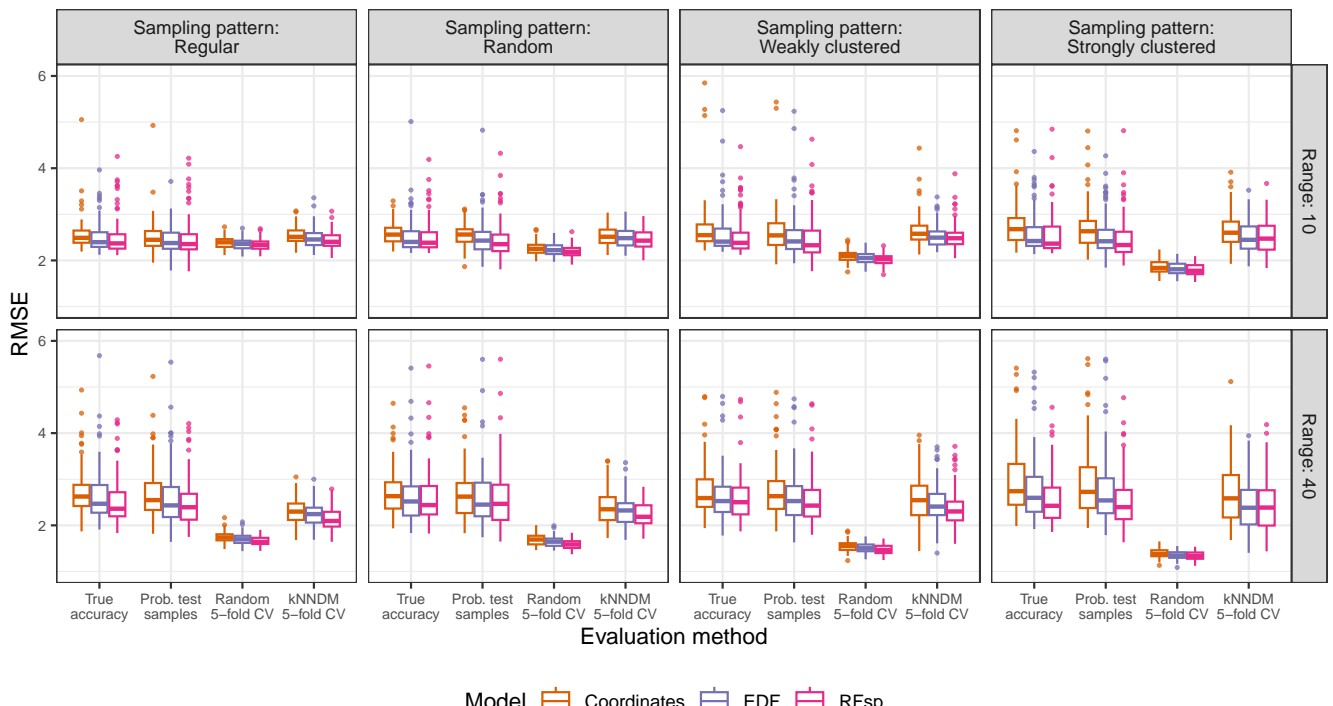

**Figure A8.** True and estimated RMSE in the interpolation area and the "complete" scenario by evaluation method, autocorrelation range, and sampling pattern. Outliers larger than 3 are not shown for visualization purposes.

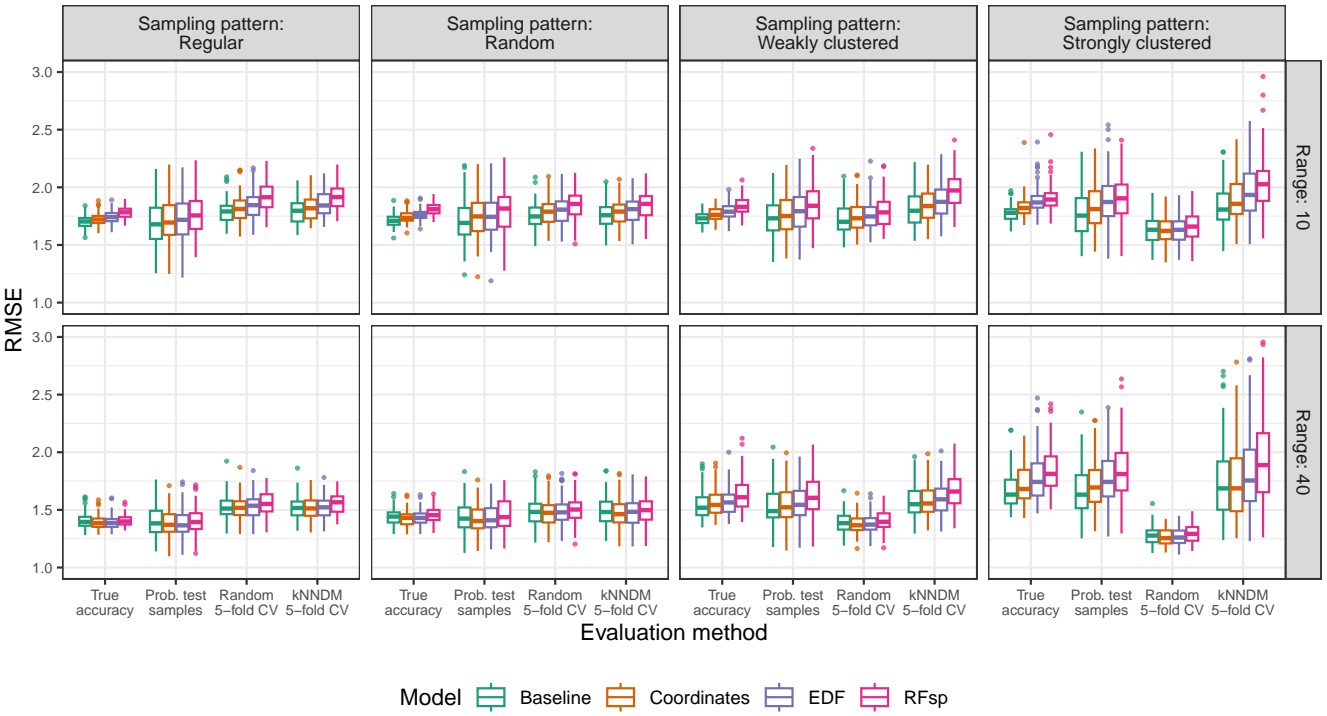

**Figure A9.** True and estimated RMSE in the interpolation area and the "missing predictors" scenario by evaluation method, autocorrelation range, and sampling pattern. Outliers larger than 3.5 are not shown for visualization purposes.

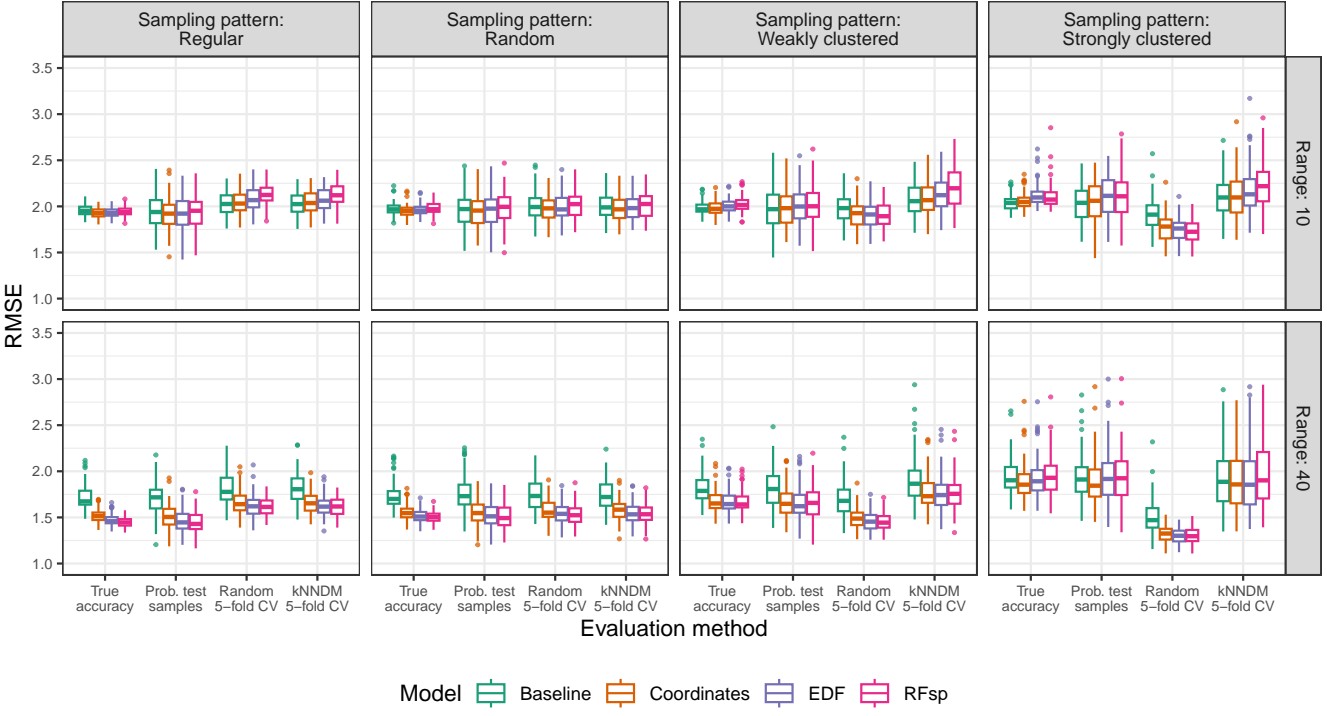

**Figure A10.** True and estimated RMSE in the interpolation area and the "proxies only" scenario by evaluation method, autocorrelation range, and sampling pattern. Results for the baseline model were not calculated as no predictors were available for modelling. Outliers larger than 4 are not shown for visualization purposes.

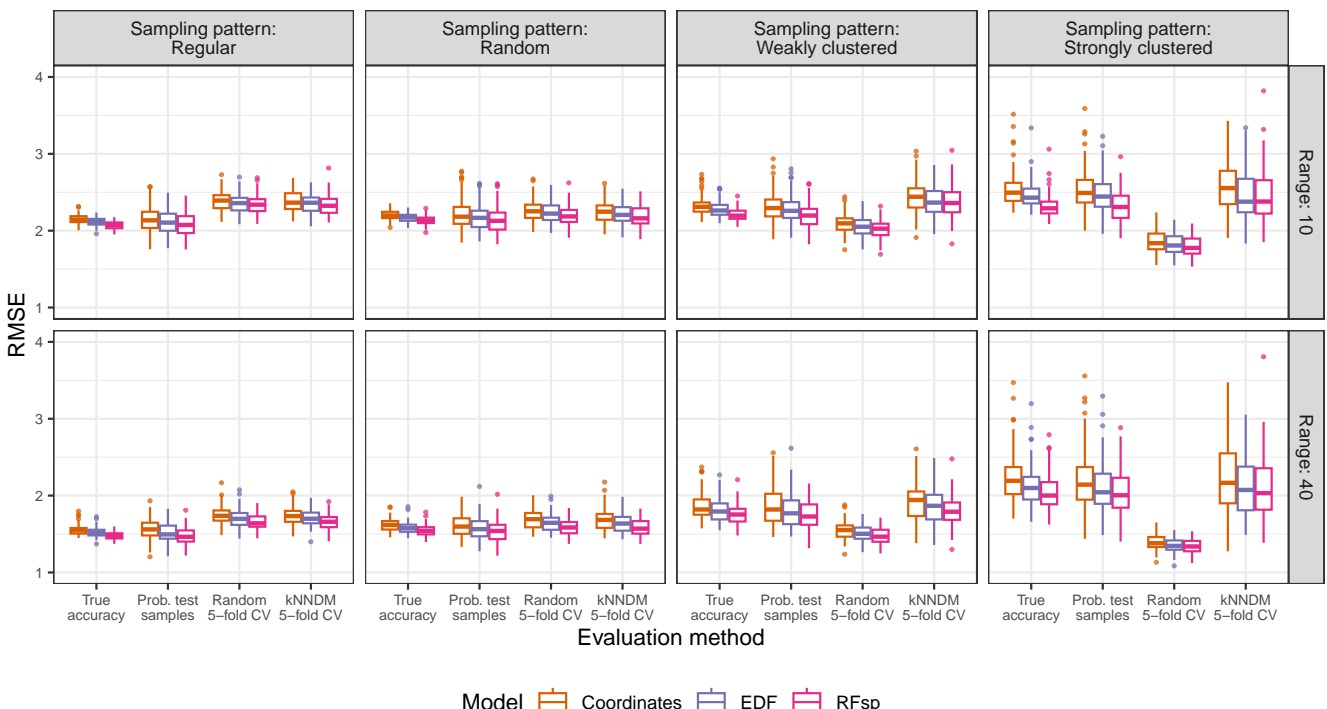

**Figure A11.** True RMSE in the extrapolation area of the best-performing standard RF for each simulation parameter combination (i.e. the standard RF model with/without proxies with the lowest median RMSE) and RF-GLS, by prediction scenario, spatial autocorrelation range, and sampling pattern.

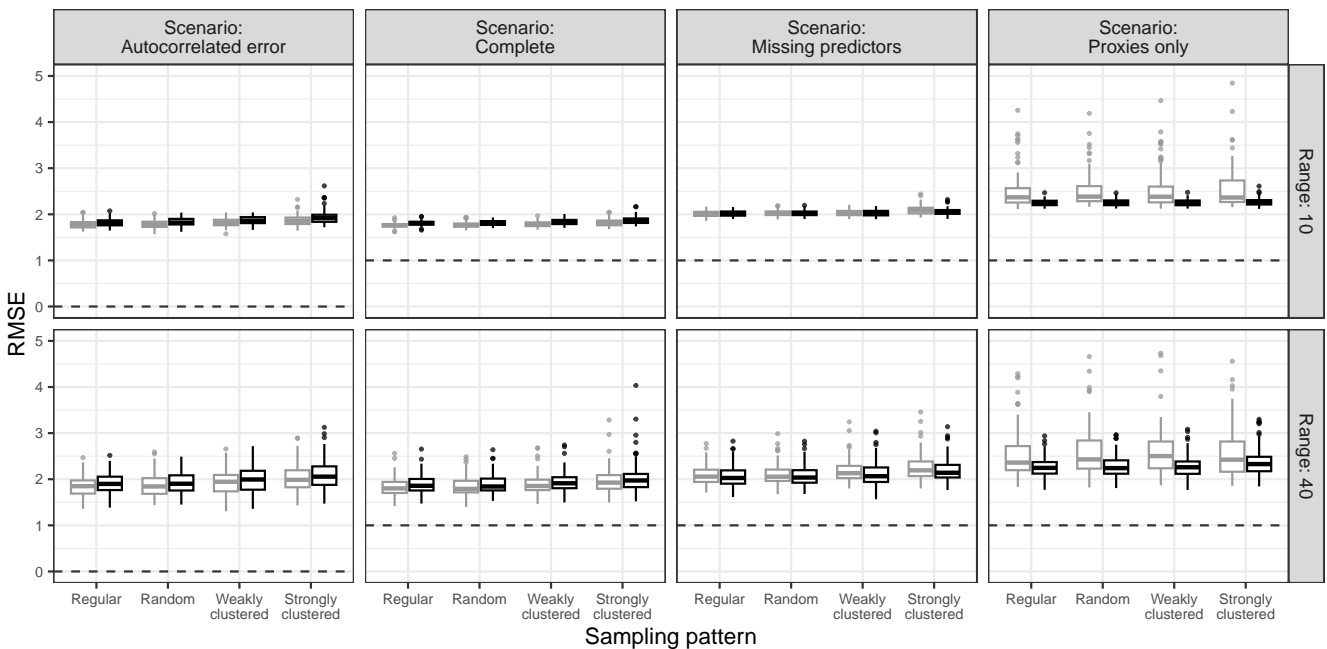

**Figure A12.** Empirical nearest neighbour distance distribution $\hat{G}$ function (A), empty space $\hat{F}$ function (B), and Ripley's $\hat{K}$ pairwise distance function (C) for the air temperature study case. The dashed red line indicates the theoretical function under complete spatial randomness (i.e. a homogeneous Poisson process) with its global envelope computed using 99 Monte Carlo simulations in grey. Empirical functions calculated from the data are in black.

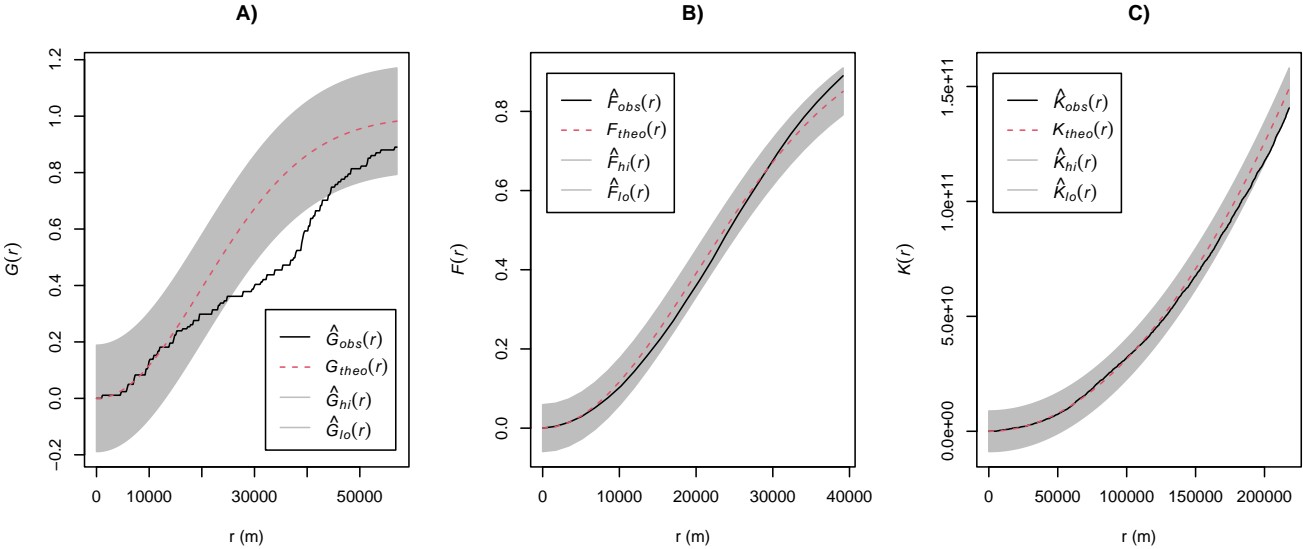

**Figure A13.** 10-fold assignment according to a random CV method (top left) and the kNNDM method (top right) for the air temperature study case. Figures at the bottom row display the corresponding Empirical Cumulative Distribution Functions (ECDF) of the geographical sample-to-sample, prediction-to-sample, and CV nearest neighbour distances. Ideally, CV-distances should match prediction-to-sample ECDF as much as possible.

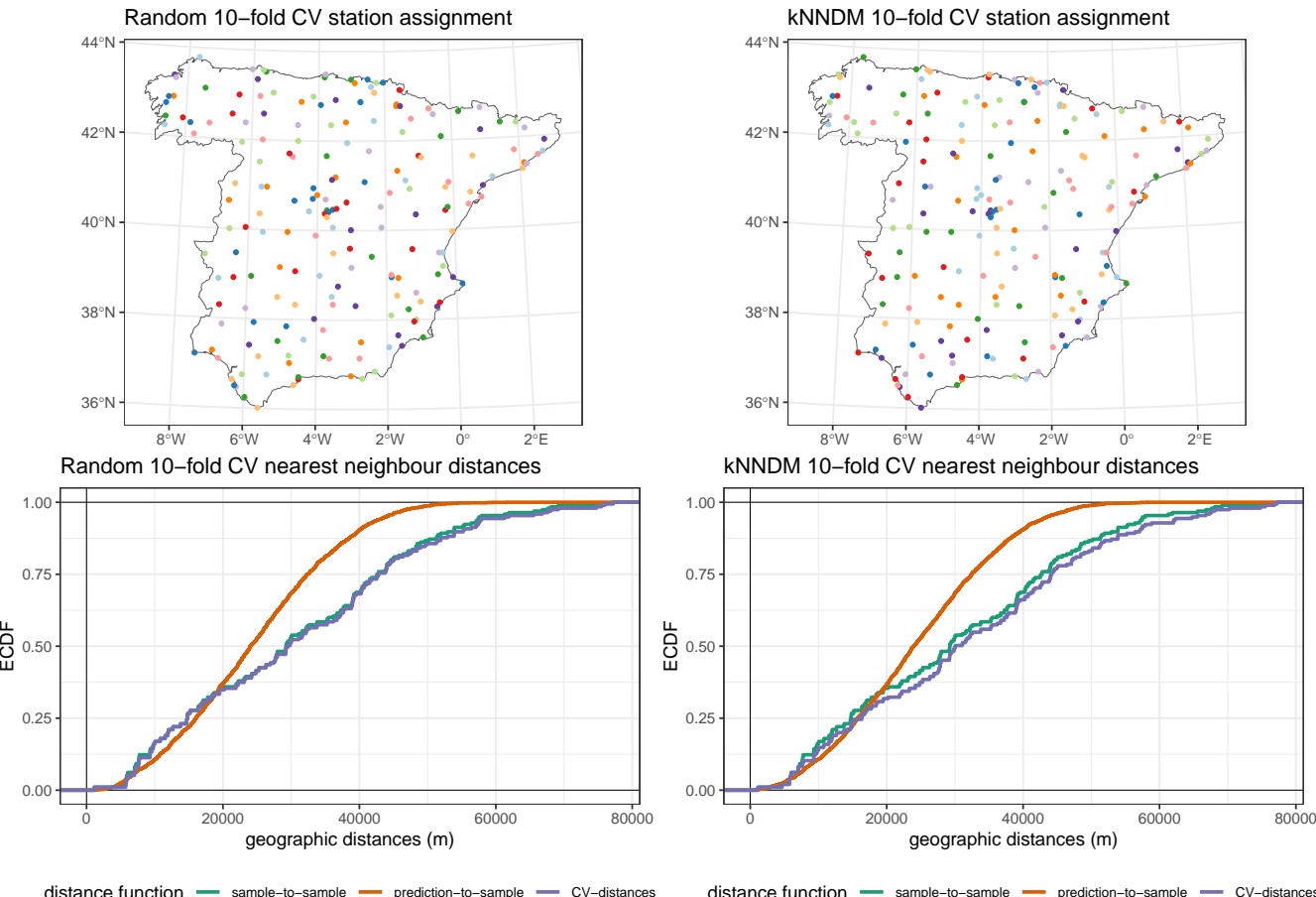

**Figure A14.** Empirical variograms for the air temperature response and residuals from all temperature models. Variogram models were fitted for illustrative purposes unless the fit did not converge.

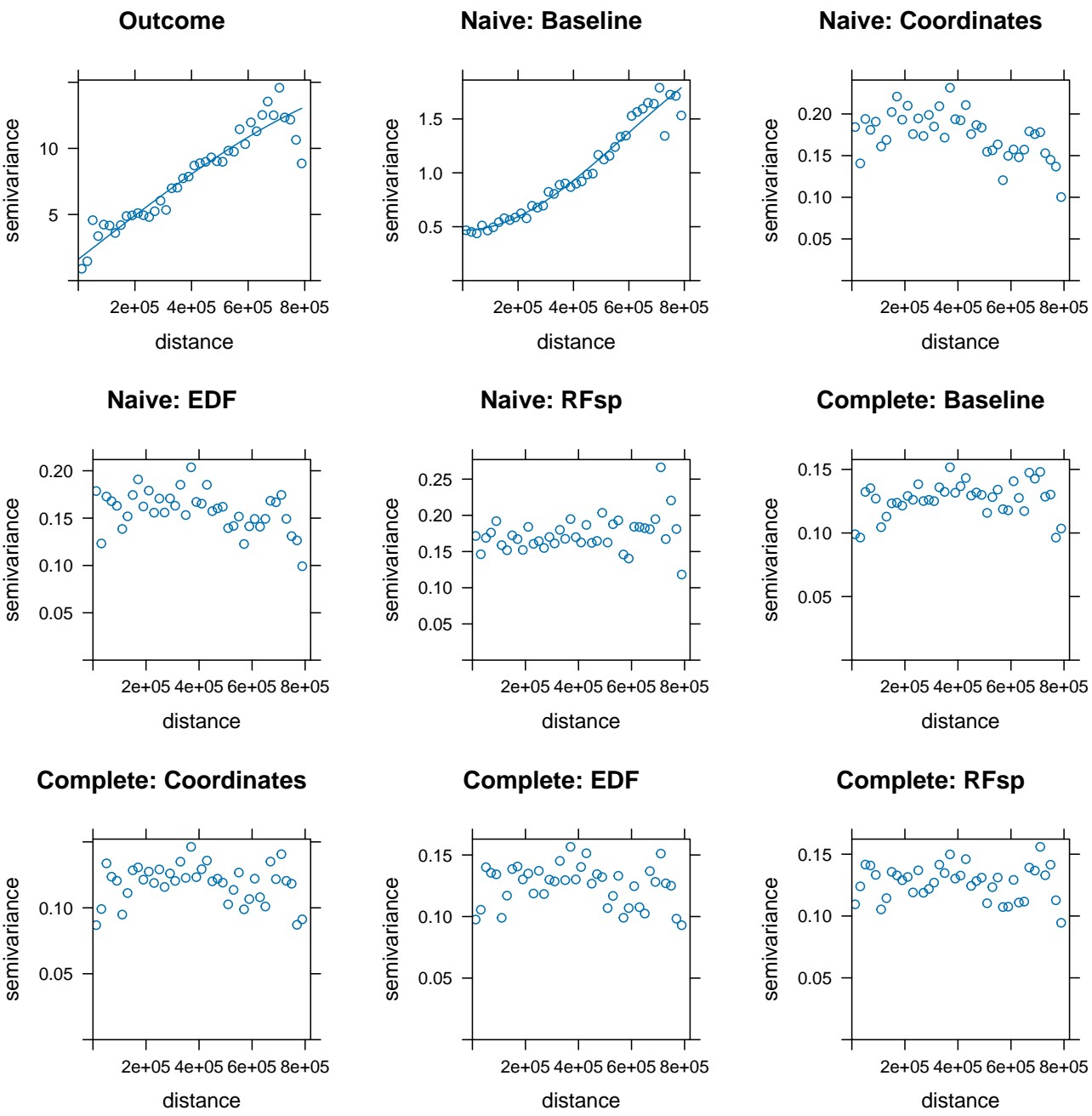

**Figure A15.** Empirical nearest neighbour distance distribution $\hat{G}$ function (A), empty space $\hat{F}$ function (B), and Ripley's $\hat{K}$ pairwise distance function (C) for the PM$_{2.5}$ study case. The dashed red line indicates the theoretical function under complete spatial randomness (i.e. a homogeneous Poisson process) with its global envelope computed using 99 Monte Carlo simulations in grey. Empirical functions calculated from the data are in black.

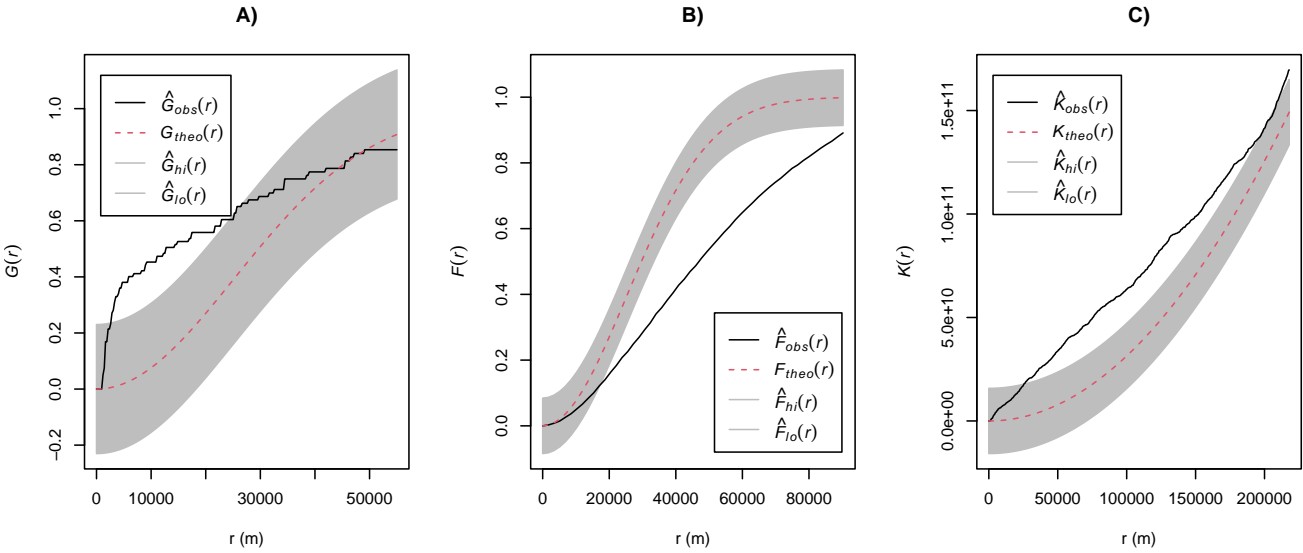

**Figure A16.** 10-fold assignment according to a random CV method (top left) and the kNNDM method (top right) for the PM$_{2.5}$ study case. Figures at the bottom row display the corresponding Empirical Cumulative Distribution Functions (ECDF) of the geographical sample-to-sample, prediction-to-sample, and CV nearest neighbour distances. Ideally, CV-distances should match prediction-to-sample ECDF as much as possible.

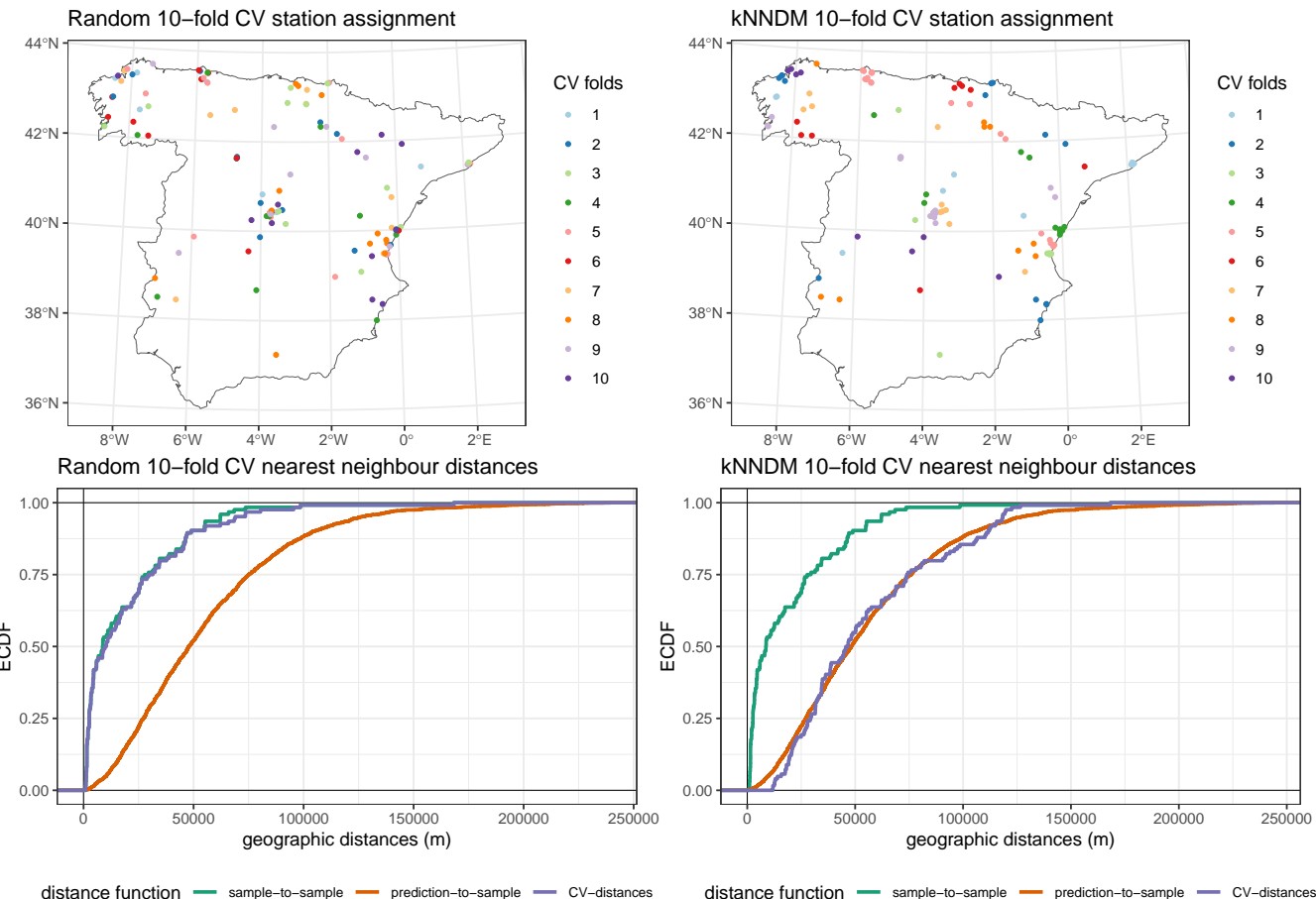

**Figure A17.** Empirical variograms for the PM$_{2.5}$ response and residuals from all PM$_{2.5}$ models. Variogram models were fitted for illustrative purposes unless the fit did not converge.

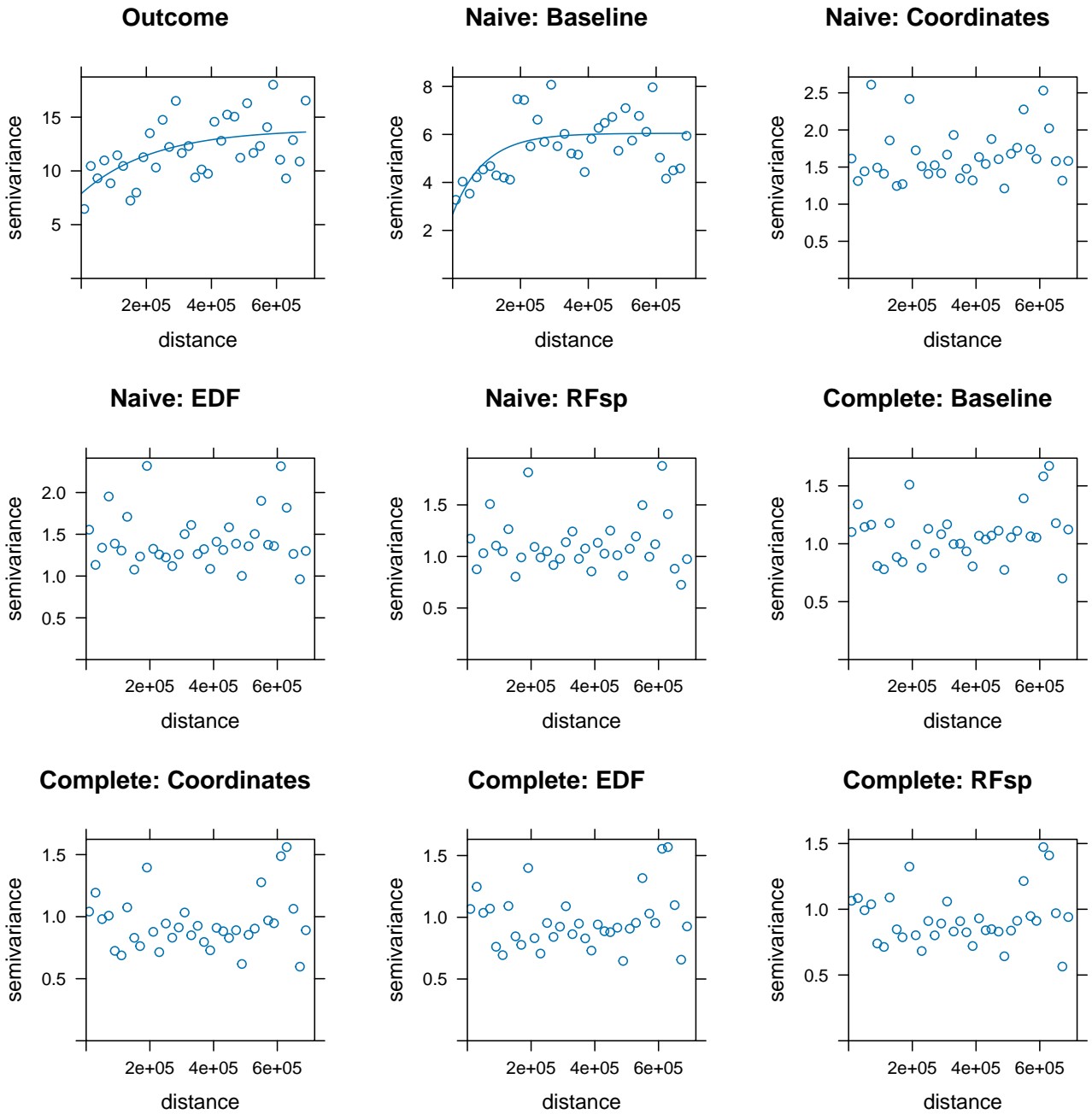

**Table A1.** List of products and their data source, original spatiotemporal resolution, and use in the complete temperature and PM$_{2.5}$ models.

| Product | Source | Original resolution | Temperature | PM$_{2.5}$ |
|---|---|---|---|---|
| Station air temperature | Agencia Estatal de Meteorología | Daily | Response | |
| Station PM$_{2.5}$ | Ministerio para la transición ecológica | Hourly/daily | | Response |
| Digital elevation model | CLMS[a]: EU-DEM v1.1 | 25 m | Predictor | Predictor |
| Distance to coast | CLMS: EU-HYDRO | Imagery interpretation | Predictor | Predictor |
| Impervious density | CLMS: IMD (2018) | 100 m | Predictor | Predictor |
| Land Cover | CLMS: CORINE Land Cover (2018) | 100 m | | Predictor |
| Population density | Eurostat: GEOSTAT (2018) | 1 km | | Predictor |
| Road density | OpenStreetMap | Imagery interpretation | | Predictor |
| NDVI (MYD13A1 v006) | MODIS Aqua Vegetation Indices | 500 m, 16-Day | Predictor | Predictor |
| Nighttime Lights | VIIRS 2019 annual VNL V2 (median) | 15", annual | | Predictor |
| PM$_{2.5}$ reanalysis | CAMS European air quality reanalysis (2019) | 0.1º, hourly | | Predictor |
| LST (MYD11A2 v006) | MODIS Aqua Land Surface Temperature | 1 km, 8-Day | Predictor | |

[a] Copernicus Land Monitoring Service.

*Author contributions.* All authors participated in the conceptualization and design of the study. CM carried out the analysis, interpreted the results, and wrote the original draft. All authors contributed to discussions, drafts, and gave final approval for publication.

*Competing interests.* The authors declare that they have no conflict of interest.

*Acknowledgements.* CM was supported by a PhD fellowship funded by the Spanish Ministerio de Ciencia e Innovación (grant reference: PRE2020-092303). We also acknowledge support from the grant CEX2018-000806-S funded by MCIN/AEI/ 10.13039/501100011033, and from the Generalitat de Catalunya through the CERCA Program.

415

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
