# Peer review of "Random forests with spatial proxies for environmental modelling: opportunities and pitfalls"

_EGUsphere, 2024_

## Author Comment (AC1)

We would like to thank the two reviewers for their time and useful feedback. In this author response, we reply to all comments and suggestions and highlight the changes we plan to implement. Our response is organised in a point-by-point fashion, where our reply follows each comment in italics. We used quotation marks for literal text mentions to our manuscript. Line numbers refer to the preprint with date 24th of January 2024.

**Anonymous Referee 1**

*This manuscript takes random forests as an example to analyze spatial agents such as coordinates and Euclidean distance fields in environmental modeling, which has positive value for spatial analysis based on machine learning models. However, there are some significant shortcomings in the work of this manuscript.*

*(1) Like other models, random forests require a set of influencing or predictive factors. Therefore, the proxy of environmental factors in spatial analysis models is not a special case of random forest models. Therefore, it is recommended that the author provide additional information on this point.*

We agree with the reviewer on the fact that the addition of spatial proxies is not a special case of a random forest, and in our manuscript we have included this information in different parts of the text.

In the introduction (lines 32-36): "The lack of consideration of space in ML models has motivated researchers to find ways to account for spatial autocorrelation to improve model performance. One straightforward approach is to add "spatial proxies" as predictors to the ML model without any modification of the algorithm. We define spatial proxies as a set of spatially-indexed variables with long or infinite autocorrelation ranges that are not causally related to the response variable".

In the methods section (Figure 1 and lines 96-99): "For each set of samples, we extracted the corresponding values of the response and predictors, deleted duplicate observations (i.e. two or more points intersecting with the same cell), and fitted a baseline model, which used predictors according to the corresponding scenario (Table 1). We also fitted coordinates, EDF, and RFsp models (see introduction for details) which included the predictors in the baseline model plus the spatial proxies".

To improve the clarity of this point, we will also include this information in the abstract by modifying the first sentence to "Spatial proxies such as coordinates and Euclidean distance fields are often added as predictors in random forest models **without any modification of the algorithm**".

*(2) The use of coordinates and Euclidean distance fields as spatial factor proxies is undoubtedly due to the influence of these spatial factors on the target, or the need to use spatial regions to reflect certain undiscovered factors. This is determined by the specific work, and such a spatial agent is undoubtedly reasonable. Even if the accuracy obtained in some models may not appear to have improved numerically. And this important aspect was not taken into account in this manuscript.*

As the reviewer, we think that one of the uses of spatial proxies is to account for residual autocorrelation prompted by the unavailability of relevant yet undiscovered or unmeasured predictors. In fact, we decided to use the word "proxy" precisely for this reason, i.e. a predictor that can be used as a substitute for these unknown or unmeasured variables. The use of proxies to account for unavailable features was also the motivation behind the "partial scenario" in the simulation study,

where only a subset of the predictors was available for modelling; as well as in the "naive models" in the study cases (lines 156-157):

"Our motivation for the naive scenario was to examine whether spatial proxies could help explain residual spatial autocorrelation due to missing predictors and therefore be used in predictor scarcity settings."

That said, we do think that this link should be made stronger throughout the text so we plan to implement the following changes. First, we will clarify the rationale behind the word "proxy" in the introduction as explained in this response. Second, we will add a new paragraph in the introduction that will discuss the possible sources of residual autocorrelation, including missing predictors and/or autocorrelated errors. Third, we will motivate the "partial scenario" and "naive models" by explicitly mentioning that the goal of these analyses is to examine the ability of coordinate and distance fields to act as proxies of the missing predictors.

*(3) It is meaningless to evaluate the superiority or inferiority of a certain agent solely based on the accuracy of the final results, without considering specific issues. In summary, it is recommended to reject the manuscript.*

As the reviewer, we also believe that accuracy is not the only way to evaluate the usefulness of a model, although in many cases where knowledge discovery is not a priority it may well be. In our discussion, we acknowledge this limitation (lines 309-314):

"Second, our analysis was based on the adequacy of spatial proxies from a prediction accuracy point of view. When using RF for knowledge discovery, variables with long or infinite autocorrelation ranges such as spatial proxies have been identified to be beyond the prediction horizon (Behrens & Viscarra Rossel 2020, Wadoux et al. 2020, Fourcade et al. 2018) and variable importance statistics in models including them should be interpreted with extreme caution (Meyer et al. 2019)."

However, when the aim is prediction as in our study, we think that accuracy metrics are indeed meaningful to evaluate the relevance of the predictors. For that purpose, we considered several specific factors including different sample distributions, autocorrelation ranges and scenarios to analyse in which cases the proxy predictors are relevant or not.

**Dr. Carsten F. Dormann**

*This study compares different approaches to address spatial autocorrelation in random forest analyses. Using simulated data, and two case studies, the authors assess prediction error and variable importance for differently clustered spatial data.*

*The study finds that clustering of spatial data had a substantial effect on RMSE, and that different spatial random forest versions differed less than that effect.*

*There are a few points I do, and a few I do not like about this study. On the plus side, I think the comparison of the RF-approaches is comprehensive and reflects nicely what people have been doing in the past. The evaluation against simulated data is how it should be (see caveat below), and I find the attempts to interpret performance using AOA nice and useful. The case studies illustrate the application case well, and also the problems, particularly using lat/lon as predictor.*

*My main criticisms are these:*

*1. The goal of the study does not become clear in the introduction and is confounded throughout the papers. To me, the structure should relate to the three "scenarios" one could have in mind for this study: interpolation, extrapolation and effect estimation (predictive inference). These targets are very different and need to be assessed differently, too. For example, regression kriging is an interpolation method (by design), while identifying importance is effect estimation. Extrapolation (to regions beyond the training data) is explicitly most often the target in the simulations presented here, but not always. I find the results hence sometimes confounding the different issues and hence have trouble interpreting them.*

*This also relates to the CV strategy. For interpolation error, random CV is fine, for extrapolation it is not. Thus, if the authors find a difference between randomCV and kNNDM-CV, this may or may not be relevant, depending on the goal of the study.*

*I think this problem pertains particularly to the introduction and the results and will not require much work to address.*

We have reviewed the introduction and confirmed that the structure from paragraph 4 onwards can be improved to better characterise the factors that determine the adequacy of spatial proxies, as well as incorporate the ideas pointed out by the reviewer. In the next version of the manuscript, we plan to restructure the second part of the introduction into three different paragraphs according to the following factors relevant for the suitability of spatial proxies:

- Study goal: interpolation vs. extrapolation vs. predictive inference (see major comment 4).

- Residual spatial autocorrelation: strength of the residual spatial autocorrelation, autocorrelation due to missing covariates vs. autocorrelated error (see major comment 2).

- Sampling pattern: regular vs. random vs. clustered samples.

These factors will also be listed in the study objectives at the end of the introduction so that they can be easily identified by the reader. In addition to these, we will also devote a separate paragraph on validation methods to empirically determine whether adding proxies is recommended (i.e. model selection), which will include probability test samples (see major comment 4) as well as random k-fold and kNNDM CV. Regarding the results section, we will also structure the simulation findings into three different blocks according to the three factors listed above.

*2. Spatial autocorrelation is entirely related to environmental variables, when in the real world it is also related to mass effects (dispersal, diffusion, contagion): the error is spatially autocorrelated, too. In my understanding, the authors did not address that. That is problematic, as Table 1's second row is thus a model WITHOUT spatial autocorrelation in the residuals, as all predictors are present to correct for it. This is, from a statistical point of view, a no-problem data set. While that does in no way invalidate the simulations, it must be clearly communicated that this is NOT a situation one would even consider using spatial representation for: no residual SA, no problem.*

*Spatial error in the residuals has been simulated in previous such studies (since Dormann et al. 2007 Ecography), and is a bit annoying to fine-tune; it can be done, though, and I think it should be done (see also simData here: https://github.com/biometry/FReibier/tree/master).*

*On the back of such simulation, a GLS fitted to the correct model would give the best possible reference analysis; anything better than that would be a biased assessment of error.*

As the reviewer pointed out, we did not consider the case of autocorrelated errors in our simulation study, where all the spatial structure in the outcome generation was introduced via covariates. To address this point, we propose to add an additional scenario "autocorrelated error" where the response generation equation includes an autocorrelated error term, for which the RF-GLS approach will probably also result to be the best approach.

Regarding the "complete" scenario (Table 1 second row) where indeed no spatial representation is needed, we discussed how spatial proxies are unnecessary in this case (lines 265-270):

"However, in the complete scenario where no residual autocorrelation was expected, we hypothesise that the similar or sometimes worse performance is due to adding an irrelevant set of predictors that are noise to the model. Unlike RK, where spatial autocorrelation is modelled in the residuals and in its absence would result in a pure nugget effect, i.e. a flat variogram leading to an ordinary least squares estimation (Hengl 2007), in a ML model the irrelevant proxies are still included in the trend model. Even though RF is fairly robust to the addition of irrelevant predictors (Kuhn & Johnson 2019), a decrease in performance was sometimes observed."

In the revised manuscript, we will also mention that no space representation is needed in this case in the methods section when the scenarios are introduced.

*3. Minimum RMSE according to simulations should be indicated in Fig. 3. Since the authors use the standard normal as error distribution, the best possible RMSE should also be 1. On average, it is closer to 2, but for the "complete, range 40" it looks as if it was below 1. That would be, well, surprising and important for interpretation: a fit into the spatial noise.*

We agree that considering the minimum possible RMSE would help to interpret results. In the revised version of the manuscript, we will add it to figure 3 by adding a dashed line. We checked our results and the minimum RMSE across all simulations is above 1 (1.04), which will become clear once we add the minimum possible RMSE horizontal line.

*4. While I read about and like the kNNDM, I still prefer a truly independent test data set. Since the authors have invented the data, they could simply extend the area by doubling it to one side, and use that second half for validation in the sense of a true extrapolation. My feeling is that kNNDM will work well if the range of data is much larger than the spatial autocorrelation, but not if SA is*

*large relative to the spatial extent. An extent of 100x100 is "only" 2.5 times larger than the range of 40. Thus, the sampled data points will fall within the SA-range virtually always (Fig. 1.2). I am not convinced that this is an independent-enough test case.*

We agree with the reviewer on that a probability test sample is always preferable to any type of cross-validation, which is something we discussed in our other works (Milà et al 2022, Linnenbrink et al 2023) but not here. In the future version of the manuscript, we will include this information in the introduction. Furthermore, we will include random probability test samples as an additional validation method in our simulations (together with the already-included random k-fold and kNNDM CV).

Regarding extrapolation, we agree on the fact that the way we designed our simulation study does not allow to isolate extrapolation effects and explore their impact on the CV results. While in some cases we will indeed have some extrapolation when predicting (e.g. with very clustered samples), there will always be interpolation as well. Following the reviewer's suggestion, we propose to modify the dimensions of the simulation area following a similar approach to our previous work (Milà et al 2022). Briefly, we will simulate in a 300x100 grid (rather than a 100x100 grid like we are doing at the moment) where samples will be taken in the 100x100 left window while the 100x100 right window will be used to evaluate extrapolation.

*5. More as a suggestion: The problem of using spatial coordinates or proxies is that they replace the causal predictors in a random forest due to collinearity. As a consequence, the importances are wrongly estimated. One can, for the simulated X, compute how well each predictor can be represented by the specific spatial predictors used. If X1 can be predicted by lat/lon or the EDFs or distances with an r of 0.8 (or so), then clearly they will compete for explanation and substantially bias importance estimates. That is the reason why the ME-approach in spdep adds the PCNMs only to explain the RESIDUALs of the model, after fitting the non-space-variables X1-X6.*

*So, I would be interested in seeing how well space can replace actual predictors. Either by reporting such "predictability of predictors by space" in the appendix, or by having another model entirely without X1-X6 in the comparison.*

*This would also tie in nicely with the difference between inter/extrapolation: space-only should work fine for inter-, but fail for extrapolation.*

Following the reviewer's recommendation, we propose to add an additional "only proxy" scenario to those listed in Table 1 where only spatial proxies will be used to predict the outcome and for which extrapolation performance should be low. Beyond this, however, we do not plan to further investigate the impact of spatial proxies on predictive inference in this specific study, as this would require a different simulation study design targeting the ability of the models to estimate the true variable importance (lines 309-314):

"Second, our analysis was based on the adequacy of spatial proxies from a prediction accuracy point of view. When using RF for knowledge discovery, variables with long or infinite autocorrelation ranges such as spatial proxies have been identified to be beyond the prediction horizon (Behrens & Viscarra Rossel 2020, Wadoux et al. 2020, Fourcade et al. 2018) and variable importance statistics in models including them should be interpreted with extreme caution (Meyer et al. 2019)."

*Overall, I think this is a nice paper almost as it is and with a little bit more integration of WHY*

*we would expect which approach to work better and a clearer structuring of the purposes of the analysis it will be just fine. IMHO it could be a greater paper, if the authors would allow for spatial autocorrelation in the error term and try to get at the bottom of WHEN space affects inference (i.e. here: importance) of predictors (that is, investigate the effect of collinearity on the intrapolation, extrapolation, variable importance).*

Thank you for the acknowledgement. As a wrap-up of the major comments, we list again the action points to address the reviewer's major comments:

- Change the structure of the introduction and results sections based on these three factors: study goal, origin and strength of the spatial autocorrelation, and sampling pattern.

- Add probability test samples as a validation method.

- Extend the area of the simulation study to 300x100 to evaluate extrapolation.

- Add the simulation "autocorrelated error" scenario to consider an autocorrelated error term.

- Add simulation "only proxy" scenario where only spatial proxy predictors are used in the models.

*Minor points: L56: Also cite other people's work here, much earlier, e.g. Le Rest et al. 2014 and whatever else we cited in Roberts et al. (2017 Ecography) on that topic.*

Since cross-validation is not the main topic of the manuscript, we kept a short reference list and did not include all references related to the topic. Nonetheless, we agree that giving more background could be useful to the non-familiar reader. We will extend the list of references for cross-validation with the works mentioned by the reviewer as well as the earlier literature regarding block-based and buffer-based spatial cross-validation.

*L57: I find the restriction to RF too narrow. This is a logical and fundamental problem, not one specific to RF. Ploton et al. (2020) showed it for random forest, Kattenborn et al. (2022) for CNNs. It is the same problem of extrapolation in space with poor design for the CV.*

Our manuscript focused on RF because it is 1) the algorithm with which spatial proxies have been mostly used, and 2) one of the most widely used models in the geosciences (lines 28-31):

"One of the most popular ML algorithms in the geospatial community is Random Forest (RF), a decision tree ensemble (Breiman 2001) that has shown good performance across many applications (Wylie et al. 2019) and centred the attention of many methodological studies (e.g. Meyer & Pebesma (2021), Hengl et al. (2018), Sekulić et al. (2020), Georganos et al. (2021), Saha et al. (2023))."

In our discussion, we acknowledged the fact that although our analysis was restricted to RF, it will possibly also apply to other algorithms as well (lines 307-309):

"First, we focused on RF regression and, while we think that our results are likely to extend to other ML algorithms, the extrapolation behaviour and sensitivity to irrelevant predictors differs by algorithm and might limit the ability to generalize our results."

That said, we agree that our work would benefit from more detail regarding the generalization of the results to other algorithms. To address this, we will include the references for other algorithms the reviewer suggested.

*L73: "scenarios" are what I called "target", "goal" or "purpose": Make clear what the goals are in the intro!*

Please see response to major comment 1 to check the new structure we propose to clarify this point.

*L178: Why would anybody use randomForest and not ranger? Much faster and hence less energy consumption.*

We used the package randomForest rather than ranger because, when running our simulations in an HPC environment, we faced threading issues when using the ranger package. While using randomForest does indeed affect energy consumption, as a different random forest implementation will not impact results. In the next round of simulations we will try to solve these issues and use ranger instead.

*What is the point of Fig. 7? I can see neither RMSEs or biases or anything, so why look at these maps? Also, we are typically more impressed by high-resolution maps, even if they are completely wrong; map visualisation is thus either uninformative and misleading in many cases.*

*What is the point of Fig. 8, apart from the funny lines in "A Coordinates"?*

We think that Figures 7 (temperature) and 8 (air pollution) are relevant since they allow us to assess 1) the differences in predicted surfaces between baseline and proxy models, and 2) whether naive models with spatial proxies can approximate the spatial patterns of complete baseline models. While in Figure 7 we showed that the inclusion of proxies in naive models resulted in a predicted surface that approximated the spatial patterns of complete models successfully, we find that this is not the case for the air pollution case study where they resulted in artifacts (Figure 8). In the revised version of the manuscript, we will further highlight these points in the text.

*Table 2 and 3: Where are the standard errors on these estimates? (Yes, I understood that some of them are a bit a pain to compute for one of the models. Still, without an estimate of the error, how can the reader interpret a value of "0.92" vs "0.87"? Might well be the same value if SD=0.4.)*

We agree with the reviewer on the importance of showing the standard deviation of the statistics to be able to ascertain whether the differences are relevant. In the future version of the manuscript, we will compute these for random k-fold cross-validation. For kNNDM, however, it will not be possible as statistics are computed, similarly to a LOO CV, by stacking all out-of-sample predicted and observed values (see lines 109-111 of the manuscript) and computing a single statistic. This is due to potential unequal sample sizes across folds, as well as how the folds are created according to the distribution of all samples simultaneously (Linnenbrink et al. 2023).

*I missed the discussion of some existing approaches to manage space into ML.*

*Hajjem, A., Bellavance, F., & Larocque, D. (2011). Mixed effects regression trees for clustered data. Statistics & Probability Letters, 81(4), 451–459. https://doi.org/10.1016/j.spl.2010.12.003*

*Hajjem, A., Bellavance, F., & Larocque, D. (2014). Mixed-effects random forest for clustered data. Journal of Statistical Computation and Simulation, 84(6), 1313–1328. https://doi.org/*

*10.1080/00949655.2012.741599*

*Li, L., Girguis, M., Lurmann, F., Wu, J., Urman, R., Rappaport, E., Ritz, B., Franklin, M., Breton, C., Gilliland, F., & Habre, R. (2019). Cluster-based bagging of constrained mixed-effects models for high spatiotemporal resolution nitrogen oxides prediction over large regions. Environment International, 128, 310–323. https://doi.org/10.1016/j.envint.2019.04.057*

*Li, L., Lurmann, F., Habre, R., Urman, R., Rappaport, E., Ritz, B., Chen, J.-C., Gilliland, F. D., & Wu, J. (2017). Constrained mixed-effect models with ensemble learning for prediction of nitrogen oxides concentrations at high spatiotemporal resolution. Environmental Science & Technology, 51(17), 9920–9929. https://doi.org/10.1021/acs.est.7b01864*

*Zhan, Y., Luo, Y., Deng, X., Chen, H., Grieneisen, M. L., Shen, X., Zhu, L., & Zhang, M. (2017). Spatiotemporal prediction of continuous daily PM2.5 concentrations across China using a spatially explicit machine learning algorithm. Atmospheric Environment, 155, 129–139. https://doi.org/10.1016/j.atmosenv.2017.02.023*

*Kattenborn, T., Schiefer, F., Frey, J., Feilhauer, H., Mahecha, M. D., & Dormann, C. F. (2022). Spatially autocorrelated training and validation samples inflate performance assessment of convolutional neural networks. ISPRS Open Journal of Photogrammetry and Remote Sensing, 5, 100018. https://doi.org/10.1016/j.ophoto.2022.100018*

*Le Rest, K., Pinaud, D., Monestiez, P., Chadoeuf, J., & Bretagnolle, V. (2014). Spatial leave-one-out cross-validation for variable selection in the presence of spatial autocorrelation. Global Ecology and Biogeography, 23, 811–820. https://doi.org/10.1111/geb.12161*

*Ploton, P., Mortier, F., Réjou-Méchain, M., Barbier, N., Picard, N., Rossi, V., Dormann, C., Cornu, G., Viennois, G., Bayol, N., Lyapustin, A., Gourlet-Fleury, S., & Pélissier, R. (2020). Spatial validation reveals poor predictive performance of large-scale ecological mapping models. Nature Communications, 11(1), Article 1. https://doi.org/10.1038/s41467-020-18321-y*

Thank you very much for providing these references; we will include them in the relevant paragraphs of the text.

**References**

Behrens, T. & Viscarra Rossel, R. A. (2020), 'On the interpretability of predictors in spatial data science: The information horizon', *Scientific Reports* **10**(1), 16737.

Breiman, L. (2001), 'Random forests', *Machine learning* **45**(1), 5–32.

Fourcade, Y., Besnard, A. G. & Secondi, J. (2018), 'Paintings predict the distribution of species, or the challenge of selecting environmental predictors and evaluation statistics', *Global Ecology and Biogeography* **27**(2), 245–256.
**URL:** *https://onlinelibrary.wiley.com/doi/abs/10.1111/geb.12684*

Georganos, S., Grippa, T., Gadiaga, A. N., Linard, C., Lennert, M., Vanhuysse, S., Mboga, N., Wolff, E. & Kalogirou, S. (2021), 'Geographical random forests: a spatial extension of the random forest algorithm to address spatial heterogeneity in remote sensing and population modelling', *Geocarto International* **36**(2), 121–136.
**URL:** *https://doi.org/10.1080/10106049.2019.1595177*

Hengl, T. (2007), 'A practical guide to geostatistical mapping of environmental variables.', *Office for Official Publications of the European Communities* .
**URL:** *https://publications.jrc.ec.europa.eu/repository/handle/JRC38153*

Hengl, T., Nussbaum, M., Wright, M. N., Heuvelink, G. B. & Gräler, B. (2018), 'Random forest as a generic framework for predictive modeling of spatial and spatio-temporal variables', *PeerJ* **6**, e5518.

Kuhn, M. & Johnson, K. (2019), *Feature engineering and selection: A practical approach for predictive models*, Chapman and Hall/CRC.

Linnenbrink, J., Milà, C., Ludwig, M. & Meyer, H. (2023), 'knndm: k-fold nearest neighbour distance matching cross-validation for map accuracy estimation', *EGUsphere* **2023**, 1–16.
**URL:** *https://egusphere.copernicus.org/preprints/2023/egusphere-2023-1308/*

Meyer, H. & Pebesma, E. (2021), 'Predicting into unknown space? estimating the area of applicability of spatial prediction models', *Methods in Ecology and Evolution* **12**(9), 1620–1633.

Meyer, H., Reudenbach, C., Wöllauer, S. & Nauss, T. (2019), 'Importance of spatial predictor variable selection in machine learning applications – moving from data reproduction to spatial prediction', *Ecological Modelling* **411**, 108815.
**URL:** *https://www.sciencedirect.com/science/article/pii/S0304380019303230*

Saha, A., Basu, S. & Datta, A. (2023), 'Random forests for spatially dependent data', *Journal of the American Statistical Association* **118**(541), 665–683.
**URL:** *https://doi.org/10.1080/01621459.2021.1950003*

Sekulić, A., Kilibarda, M., Heuvelink, G. B., Nikolić, M. & Bajat, B. (2020), 'Random forest spatial interpolation', *Remote Sensing* **12**(10).
**URL:** *https://www.mdpi.com/2072-4292/12/10/1687*

Wadoux, A. M. J.-C., Samuel-Rosa, A., Poggio, L. & Mulder, V. L. (2020), 'A note on knowledge discovery and machine learning in digital soil mapping', *European Journal of Soil Science* **71**(2), 133–136.
**URL:** *https://bsssjournals.onlinelibrary.wiley.com/doi/abs/10.1111/ejss.12909*

Wylie, B. K., Pastick, N. J., Picotte, J. J. & Deering, C. A. (2019), 'Geospatial data mining for digital raster mapping', *GIScience & Remote Sensing* **56**(3), 406–429.
**URL:** *https://www.tandfonline.com/doi/abs/10.1080/15481603.2018.1517445*

---

## Author Response (AR2)

In this document we reply to all comments and suggestions made on our original manuscript submitted on the 24th of January, and list the changes we have implemented in this revised version. The document is organised by reviewer, where our reply follows each comment in italics. The included line numbers, text, and figures correspond to the revised version of the manuscript.

**Anonymous Referee 1**

*This manuscript takes random forests as an example to analyze spatial agents such as coordinates and Euclidean distance fields in environmental modeling, which has positive value for spatial analysis based on machine learning models. However, there are some significant shortcomings in the work of this manuscript.*

*(1) Like other models, random forests require a set of influencing or predictive factors. Therefore, the proxy of environmental factors in spatial analysis models is not a special case of random forest models. Therefore, it is recommended that the author provide additional information on this point.*

We agree with the reviewer on the fact that the addition of spatial proxies as predictors is not a special case of a random forest, and in our manuscript we have included this information in the introduction (lines 40-42):

"The lack of consideration of space in ML models has motivated researchers to try to find ways to account for spatial autocorrelation to improve model performance. One straightforward approach is to add "spatial proxies" as predictors to the ML model without any modification of the algorithm."

In the revised version of the manuscript, we now also include this information in the abstract (lines 1-3):

"Spatial proxies such as coordinates and distance fields are often added as predictors in Random Forest (RF) models without any modification of the algorithm to account for residual autocorrelation and improve predictions; however, their suitability under different predictive conditions encountered in environmental applications has not yet been assessed."

*(2) The use of coordinates and Euclidean distance fields as spatial factor proxies is undoubtedly due to the influence of these spatial factors on the target, or the need to use spatial regions to reflect certain undiscovered factors. This is determined by the specific work, and such a spatial agent is undoubtedly reasonable. Even if the accuracy obtained in some models may not appear to have improved numerically. And this important aspect was not taken into account in this manuscript.*

As the reviewer, we think that one of the uses of spatial proxies is to account for residual autocorrelation prompted by the unavailability of relevant yet undiscovered or unmeasured predictors. In fact, we decided to use the word "proxy" precisely for this reason, i.e. a predictor that can be be used as a surrogate for unobserved factors that can cause residual spatial autocorrelation. In order to clarify this point, we now include the rationale behind the word "proxy" in the introduction (lines 42-45):

"We define spatial proxies as a set of spatially-indexed variables with long or infinite autocorrelation ranges that are not causally related to the response. We use the term "proxy" since these predictors act as surrogates for unobserved factors, such as missing predictors or an autocorrelated error term, that can cause residual autocorrelation."

Second, we have added a new paragraph in the introduction that discusses the possible sources of residual autocorrelation, including missing predictors and/or autocorrelated errors (lines 72-73):

"The second factor is residual autocorrelation, which typically arises when a relevant predictor is not available for modelling because it is either unmeasured or unknown, or because the error term is autocorrelated (F. Dormann et al. 2007)."

Third, the situation described by the reviewer is contemplated in our analyses in the "missing predictors" simulation scenario, which we now motivate (lines 135-139):

"We generated predictor and response surfaces (Fig. 1.1) according to the different scenarios described in Table 1: 1) "autocorrelated error", where residual autocorrelation is expected due to a spatially autocorrelated error term; 2) "complete", where no spatial autocorrelation is expected and therefore spatial proxies are assumed to be irrelevant; 3) "missing predictors", where residual autocorrelation is present due to missing predictors; and finally 4) "proxies only", where no predictors are available for modelling and only proxies are used."

In this case, the addition of proxies can be beneficial for spatial interpolation as the reviewer points out (lines 252-253):

"On the other hand, in scenarios where residual autocorrelation was expected either due to an autocorrelated error term or missing predictors, models with spatial proxies showed smaller errors in many instances."

We also motivate naive models in the two case studies by saying that spatial autocorrelation is due to missing predictors (lines 225-226):

"Our motivation for the naive model was to examine whether spatial proxies could help explaining residual spatial autocorrelation due to missing predictors and therefore be used in predictor scarcity settings."

Finally, for the temperature study case, we found positive effects of using spatial proxies since the conditions in terms of residual spatial autocorrelation and sample distribution were appropriate (lines 370-372):

"We found that a model with only a DEM and spatial proxies managed to account for the residual spatial autocorrelation, and performed almost as well as a much more comprehensive model which produced similar predicted surfaces. This highlights the value of spatial proxies for cost-effective predictive modelling as long as the conditions outlined above are met."

*(3) It is meaningless to evaluate the superiority or inferiority of a certain agent solely based on the accuracy of the final results, without considering specific issues. In summary, it is recommended to reject the manuscript.*

As the reviewer, we also believe that accuracy is not the only way to evaluate the usefulness of a model, although in many cases where knowledge discovery is not a priority it may well be. In our discussion, we acknowledge this point (lines 383-387):

"Second, our analysis was based on the adequacy of spatial proxies from a prediction accuracy point of view. When using RF for knowledge discovery, variables with long or infinite autocorrelation ranges such as spatial proxies have been identified to be beyond the prediction horizon (Behrens & Viscarra Rossel 2020, Wadoux, Samuel-Rosa, Poggio & Mulder 2020, Fourcade et al. 2018) and variable importance statistics in models including them should be interpreted with extreme caution (Meyer et al. 2019, Wadoux, Minasny & McBratney 2020)."

However, when the aim is prediction as in our study, we think that accuracy metrics are indeed meaningful to evaluate the relevance of the predictors. For that purpose, we considered several specific factors including different sampling patterns, autocorrelation ranges, and simulation scenarios to analyse in which cases the proxy predictors are relevant.

**Dr. Carsten F. Dormann**

*This study compares different approaches to address spatial autocorrelation in random forest analyses. Using simulated data, and two case studies, the authors assess prediction error and variable importance for differently clustered spatial data.*

*The study finds that clustering of spatial data had a substantial effect on RMSE, and that different spatial random forest versions differed less than that effect.*

*There are a few points I do, and a few I do not like about this study. On the plus side, I think the comparison of the RF-approaches is comprehensive and reflects nicely what people have been doing in the past. The evaluation against simulated data is how it should be (see caveat below), and I find the attempts to interpret performance using AOA nice and useful. The case studies illustrate the application case well, and also the problems, particularly using lat/lon as predictor.*

*My main criticisms are these:*

*1. The goal of the study does not become clear in the introduction and is confounded throughout the papers. To me, the structure should relate to the three "scenarios" one could have in mind for this study: interpolation, extrapolation and effect estimation (predictive inference). These targets are very different and need to be assessed differently, too. For example, regression kriging is an interpolation method (by design), while identifying importance is effect estimation. Extrapolation (to regions beyond the training data) is explicitly most often the target in the simulations presented here, but not always. I find the results hence sometimes confounding the different issues and hence have trouble interpreting them.*

*This also relates to the CV strategy. For interpolation error, random CV is fine, for extrapolation it is not. Thus, if the authors find a difference between randomCV and kNNDM-CV, this may or may not be relevant, depending on the goal of the study.*

*I think this problem pertains particularly to the introduction and the results and will not require much work to address.*

We reviewed the structure of the manuscript and confirmed that it can be improved to enhance the narrative and avoid confusion of the different elements we investigated. The first key element we changed are the objectives, which we have now structured in three main points and one additional secondary objective (lines 118-127):

"In this work, we investigate several RF models with spatial proxies, namely coordinates, EDF, and RFsp, with the following objectives:

1. To assess the suitability of spatial proxies depending on different factors: the modelling objective (interpolation vs. extrapolation), the strength of the residual spatial autocorrelation, and the sampling pattern.
2. To investigate which validation methods can be used as a model selection tool to empirically assess the suitability of spatial proxies and select the most appropriate proxy configuration.
3. To provide guidance to practitioners regarding the use of spatial proxies in real-world applications.

We address the first two objectives in a simulation study, while for the third objective we carry out two case studies where we model air temperature and particulate air pollution in Spain. We further compare and discuss the findings in the context of the recently developed RF-GLS model to benchmark the performance of this alternative modelling approach."

The new objectives now drive the structure of the abstract, introduction, results, and discussion sections. In the abstract, the new objectives are listed (lines 3-7):

"We investigate 1) the suitability of spatial proxies depending on the modelling objective (interpolation vs. extrapolation), the strength of the residual spatial autocorrelation, and the sampling pattern; 2) which validation methods can be used as a model selection tool to empirically assess the suitability of spatial proxies; and show 3) the effect of using spatial proxies in real-world environmental applications."

In the introduction section, we have restructured the text according to the new objective list (we do not include all text here because it comprises most of the introduction):

- Paragraphs 4-7 address the first objective as follows: paragraph 4 presents the factors that can affect the suitability of spatial proxies (lines 55-58), paragraph 5 addresses the first factor, the model's objective (lines 59-71), paragraph 6 addresses the second factor, the origin and strength of the spatial autocorrelation (lines 72-80), and paragraph 7 addresses the third factor, the sampling pattern (lines 81-89).
- Paragraph 8 introduces objective 2, i.e. validation methods for model selection (lines 90-105).
- Paragraph 9 addresses the additional objective by listing alternative models to RF with spatial proxies including RF-GLS (lines 106-115).

In the results section, we now use subsection indicators that directly map to the study objectives:

- Section "3.1.1 Suitability of spatial proxies" refers to the first objective, where now the results of each factor (models' objective, autocorrelation, sampling pattern) are listed one at a time in separate paragraphs.
- Section "3.1.2 Validation methods for proxy selection" refers to the second objective.
- Section "3.1.3 Comparison of spatial proxies with RF-GLS" refers to the additional objective.
- Section "3.2 Case studies" refers to the third objective.

Finally, in the discussion section, we have structured the text as follows:

- Paragraphs 1-3 discuss the first objective regarding the suitability of spatial proxies, where paragraph 1 is devoted to the model's objective (lines 314-322), paragraph 2 delves into spatial autocorrelation (lines 323-336), and paragraph 3 discusses the effect of the sampling pattern (lines 337-349).
- Paragraph 4 discusses the second objective regarding the validation methods for proxy variable selection (lines 350-358).
- Paragraph 5 discusses the additional objective regarding the performance of RF-GLS (lines 359-367).
- Paragraph 6 discusses the third objective regarding the real-world examples (lines 368-379).

*2. Spatial autocorrelation is entirely related to environmental variables, when in the real world it is also related to mass effects (dispersal, diffusion, contagion): the error is spatially autocorrelated, too. In my understanding, the authors did not address that. That is problematic, as Table 1's second row is thus a model WITHOUT spatial autocorrelation in the residuals, as all predictors are present to correct for it. This is, from a statistical point of view, a no-problem data set. While that does in no way invalidate the simulations, it must be clearly communicated that this is NOT a situation one would even consider using spatial representation for: no residual SA, no problem.*

*Spatial error in the residuals has been simulated in previous such studies (since Dormann et al. 2007 Ecography), and is a bit annoying to fine-tune; it can be done, though, and I think it should*

*be done (see also simData here: https://github.com/biometry/FReibier/tree/master).*

*On the back of such simulation, a GLS fitted to the correct model would give the best possible reference analysis; anything better than that would be a biased assessment of error.*

As the reviewer points out, we did not consider the case of autocorrelated errors in our simulation study, where all the spatial structure in the outcome generation was introduced via covariates. To address this point, we have added an additional scenario "autocorrelated error" where the response generation equation includes an autocorrelated error term (lines 135-136):

"We generated predictor and response surfaces (Fig. 1.1) according to the different scenarios described in Table 1: 1) "autocorrelated error", where residual autocorrelation is expected due to a spatially autocorrelated error term"

The results of this new scenario are available in the revised manuscript and included in all relevant figures, one of which we include in this report (report figure 1).

[Figure]

Figure 1: True RMSE in the interpolation area of each model type by scenario, autocorrelation range, and sampling pattern. The dashed line indicates the minimum possible RMSE for each scenario. RMSE for the baseline model in the "proxies only" scenario uses a constant prediction calculated as the average response value in the training data. Outliers larger than 3 are not shown for visualization purposes.

With these new results, we can confirm the intuition of the reviewer that a GLS-like model such as RF-GLS provides the best results in the new scenario (lines 284-287):

"RF-GLS outperformed or was on a par with the best-performing standard RF model with and without proxies for all parameter combinations in both the interpolation (Fig. 7) and extrapolation (supporting Fig. 11) areas in the simulation study. The most relevant gains in performance when comparing RF-GLS to RF with and without proxies were in the "autocorrelated error" scenario for the interpolation area with regular and random samples, for which RMSE were substantially lower."

Regarding the "complete" scenario where indeed no spatial representation is needed, we now high-light that spatial proxies are unnecessary in the simulation methods section (lines 136-137):

"2) "complete", where no spatial autocorrelation is expected and therefore spatial proxies are assumed to be irrelevant"

We discuss it in more detail it in the discussion section (lines 328-333):

"However, in complete models with no residual autocorrelation, the similar or sometimes worse performance is due to adding an irrelevant set of predictors that are noise to the model. Unlike regression kriging, where spatial autocorrelation is modelled in the residuals and in its absence would result in a pure nugget effect, i.e. a flat variogram leading to an ordinary least squares estimation (Hengl 2007), in a ML model the irrelevant proxies are still included. Even though RF is fairly robust to the addition of irrelevant predictors (Kuhn & Johnson 2019), a decrease in performance was sometimes observed."

*3. Minimum RMSE according to simulations should be indicated in Fig. 3. Since the authors use the standard normal as error distribution, the best possible RMSE should also be 1. On average, it is closer to 2, but for the "complete, range 40" it looks as if it was below 1. That would be, well, surprising and important for interpretation: a fit into the spatial noise.*

We have added a dashed horizontal line indicating the minimum RMSE by scenario in all relevant figures, see for example report Figure 1. We explain the origin of the expected minimum RMSE in the methods section (lines 159-161):

"The expected minimum possible RMSE for scenarios 2-4 was equal to 1 (standard deviation of the random error), whereas it was equal to 0 for scenario "autocorrelated error" as the error could potentially be explained by the proxies."

*4. While I read about and like the kNNDM, I still prefer a truly independent test data set. Since the authors have invented the data, they could simply extend the area by doubling it to one side, and use that second half for validation in the sense of a true extrapolation. My feeling is that kNNDM will work well if the range of data is much larger than the spatial autocorrelation, but not if SA is large relative to the spatial extent. An extent of 100x100 is "only" 2.5 times larger than the range of 40. Thus, the sampled data points will fall within the SA-range virtually always (Fig. 1.2). I am not convinced that this is an independent-enough test case.*

We thank the reviewer for his suggestions. We have expanded our simulation study to now include both the extrapolation model objective, as well as probability test samples (see the updated simulation workflow in report Figure 2).

Regarding the independent test data, we agree with the reviewer on that an independent test dataset is always preferable to any type of cross-validation, which is something we discussed in our other works (Milà et al 2022, Linnenbrink et al 2023) but not here. In the revised version of the manuscript, we have included this information in the introduction (lines 93-96):

"Amongst validation methods, probability test sampling is the preferred approach as it offers unbiased estimates (Wadoux et al. 2021) that can be used for model selection. Unfortunately, independent test samples are rarely available in the field of environmental sciences, and alternative validation methods such as Cross-Validation (CV) must be used."

Furthermore, we have included random probability test samples as an additional validation method in our simulations (lines 162-164):

[Figure]

Figure 2: Workflow of the simulation study.

"Since the true RMSE is unknown in real-world applications, we also estimated the RMSE using additional validation methods (Fig. 1.5). First, a probability sample of 100 random test points was drawn and used to estimate the RMSE in the interpolation and extrapolation areas separately."

Results for the new accuracy estimation method have been included in the relevant sections, see for example report Figure 3. As expected, results for probability test samples indicate that it is a good method to estimate performance as well as for model selection both in the interpolation and extrapolation areas (lines 351-355):

"we showed that random CV underestimates map accuracy when assessing extrapolation performance or interpolation with clustered samples, which has been shown before (Linnenbrink et al. 2023, Wadoux et al. 2021). Perhaps even more important, random CV incorrectly ranks models in those instances, systematically favouring models with proxies even though those are not always appropriate. On the other hand, probability test samples and kNNDM CV did provide correct model ranks."

Regarding extrapolation, we agree on the fact that the way we designed our simulation study did not allow to isolate extrapolation effects and explore their impact on the CV results. While in

[Figure]

Figure 3: True and estimated RMSE in the interpolation area and the "autocorrelated error" scenario by evaluation method, autocorrelation range, and sampling pattern. Values larger than 3.5 are not shown for visualization purposes.

some cases we will indeed have some extrapolation when predicting (e.g. with clustered samples), there will always be interpolation as well. Following the reviewer's suggestion, we have modified the dimensions of the simulation area following a similar approach to our previous work (Milà et al. 2022). Briefly, we have simulated in a 300x100 grid where samples are drawn in the 100x100 left window while the 100x100 right window is used to evaluate extrapolation (lines 130-133):

"We designed a simulation study on a virtual 300x100 grid to assess, in different prediction settings, the suitability of RF regression models using three different types of spatial proxies: coordinates, EDF, and RFsp (Fig. 1). Within the grid, two separate areas were defined (Fig. 1.1): sampling, from where observations were sampled and which coincided with the interpolation prediction area; and the extrapolation prediction area, used to evaluate spatial model transferability."

New results for extrapolation indicate that, as expected, RF with spatial proxies are always detrimental (report Figure 4).

*5. More as a suggestion: The problem of using spatial coordinates or proxies is that they replace the causal predictors in a random forest due to collinearity. As a consequence, the importances are wrongly estimated. One can, for the simulated X, compute how well each predictor can be represented by the specific spatial predictors used. If X1 can be predicted by lat/lon or the EDFs or distances with an r of 0.8 (or so), then clearly they will compete for explanation and substantially bias importance estimates. That is the reason why the ME-approach in spdep adds the PCNMs only to explain the RESIDUALs of the model, after fitting the non-space-variables X1-X6.*

*So, I would be interested in seeing how well space can replace actual predictors. Either by reporting such "predictability of predictors by space" in the appendix, or by having another model entirely*

[Figure]

Figure 4: True RMSE in the extrapolation area of each model type by scenario, autocorrelation range, and sampling pattern. The dashed line indicates the minimum possible RMSE for each scenario. RMSE for the baseline model in the "proxies only" scenario uses a constant prediction calculated as the average response value in the training data. Outliers larger than 5 are not shown for visualization purposes.

*without X1-X6 in the comparison.*

*This would also tie in nicely with the difference between inter/extrapolation: space-only should work fine for inter-, but fail for extrapolation.*

Following the reviewer's recommendation, we have added the additional "proxies only" scenario where only spatial proxies are used to predict the outcome (lines 135-139):

"We generated predictor and response surfaces (Fig. 1.1) according to the different scenarios described in Table 1: 1) "autocorrelated error", where residual autocorrelation is expected due to a spatially autocorrelated error term; 2) "complete", where no spatial autocorrelation is expected and therefore spatial proxies are assumed to be irrelevant; 3) "missing predictors", where residual autocorrelation is present due to missing predictors; and finally 4) "proxies only", where no predictors are available for modelling and only proxies are used."

In this new analysis, we confirmed the reviewer's intuition that only using spatial proxies for extrapolation is a very bad idea, as it results in larger errors compared to a constant mean prediction (see report Figure 4), whereas for interpolation it may actually perform almost as well as a model with the causal predictors (lines 264-266 and report figure 1):

"Finally, interpolation models using only spatial proxies as predictors performed nearly as well as models with all (scenario: complete) or a subset (scenario: missing predictors) of predictors provided samples were regularly or randomly distributed and the autocorrelation range was 40 (Fig. 4)."

Beyond this, however, we have not further investigated the impact of spatial proxies on predictive inference, as we think this would require a different simulation study design targeting the ability

of the models to estimate the true variable importance as acknowledged in the discussion section (lines 383-387):

"Second, our analysis was based on the adequacy of spatial proxies from a prediction accuracy point of view. When using RF for knowledge discovery, variables with long or infinite autocorrelation ranges such as spatial proxies have been identified to be beyond the prediction horizon (Behrens & Viscarra Rossel 2020, Wadoux, Samuel-Rosa, Poggio & Mulder 2020, Fourcade et al. 2018) and variable importance statistics in models including them should be interpreted with extreme caution (Meyer et al. 2019, Wadoux, Minasny & McBratney 2020)."

*Overall, I think this is a nice paper almost as it is and with a little bit more integration of WHY we would expect which approach to work better and a clearer structuring of the purposes of the analysis it will be just fine. IMHO it could be a greater paper, if the authors would allow for spatial autocorrelation in the error term and try to get at the bottom of WHEN space affects inference (i.e. here: importance) of predictors (that is, investigate the effect of collinearity on the intrapolation, extrapolation, variable importance).*

We thank the reviewer for the acknowledgement. As a wrap-up of the major comments, we list again the action points we have taken to address the reviewer's concerns:

- We have changed the study objectives to improve the narrative of the study.

- We have changed the structure of the abstract, introduction, results, and discussion sections based on the new objectives so that the different elements do not get confused.

- We have added probability test samples as a validation method.

- We have extended the area of the simulation study to 300x100 and evaluated extrapolation.

- We have added the simulation scenario "autocorrelated error".

- We have added the simulation scenario "proxies only".

*Minor points: L56: Also cite other people's work here, much earlier, e.g. Le Rest et al. 2014 and whatever else we cited in Roberts et al. (2017 Ecography) on that topic.*

We have extended the list of references for cross-validation with the works mentioned by the reviewer as well as the earlier literature regarding block-based and buffer-based spatial cross-validation (lines 99-103):

"Several spatial CV methods have been proposed to address the limitations of standard validation approaches (Roberts et al. 2017, Ploton et al. 2020, Kattenborn et al. 2022) using CV based on spatial blocking (Wenger & Olden 2012, Valavi et al. 2019), buffering (Telford & Birks 2009, Le Rest et al. 2014), clustering (Wang et al. 2023), as well as sampling-intensity weighted CV and model-based geostatistical approaches (de Bruin et al. 2022)."

*L57: I find the restriction to RF too narrow. This is a logical and fundamental problem, not one specific to RF. Ploton et al. (2020) showed it for random forest, Kattenborn et al. (2022) for CNNs. It is the same problem of extrapolation in space with poor design for the CV.*

In our new version of the introduction, the accuracy estimation text is now more general and does not only refer specifically to random forest (lines 90-105). That said, our manuscript focused on

RF because it is 1) the algorithm with which spatial proxies have been mostly used, and 2) one of the most widely used models in the geosciences (lines 36-39):

"One of the most popular ML algorithms in the geospatial community is Random Forest (RF), a decision tree ensemble (Breiman 2001) that has shown good performance across many applications (Wylie et al. 2019) and centred the attention of many methodological studies (e.g. Meyer & Pebesma 2021, Hengl et al. 2018, Sekulić et al. 2020, Georganos et al. 2021, Saha et al. 2023)."

In the discussion section, we acknowledged the fact that although our analysis was restricted to RF, it will possibly also apply to other algorithms as well (lines 381-383):

"First, we focused on RF regression and, while we think that our results likely extend to other ML algorithms, the extrapolation behaviour and sensitivity to irrelevant predictors differs by algorithm and might limit the ability to generalize our results."

*L73: "scenarios" are what I called "target", "goal" or "purpose": Make clear what the goals are in the intro!*

Please see response to major comment 1 to check the new structure we implemented to clarify this point.

*L178: Why would anybody use randomForest and not ranger? Much faster and hence less energy consumption.*

We used the package randomForest rather than ranger because, when running our simulations in an HPC environment, we faced threading issues when using the ranger package. In this new version, however, we have been able to solve them and use ranger instead (lines 336-240):

"Our analyses were carried out in R version 4.2.2 (R Core Team 2022) using several packages: `sf` (Pebesma 2018) and `terra` (Hijmans 2022) for spatial data management; `caret` (Kuhn 2022), `ranger` (Wright & Ziegler 2017), `RandomForestsGLS` (Saha et al. 2022), and `CAST` (Meyer et al. 2023) for spatial modelling; `gstat` (Pebesma 2004) for random field simulation; and `ggplot2` (Wickham 2016) and `tmap` (Tennekes 2018) for graphics and cartographic representations. Additional packages were used for other minor tasks."

*What is the point of Fig. 7? I can see neither RMSEs or biases or anything, so why look at these maps? Also, we are typically more impressed by high-resolution maps, even if they are completely wrong; map visualisation is thus either uninformative and misleading in many cases.*

*What is the point of Fig. 8, apart from the funny lines in "A Coordinates"?*

We think that the figures mentioned by the reviewer are relevant since they allow us to assess 1) the differences in predicted surfaces between baseline and proxy models, 2) the sensitivity to the different proxies used, and 3) whether naive models with spatial proxies can approximate the spatial patterns of complete models. To highlight these, we have extended the text regarding those points in the results section (lines 296-299 and 306-310):

"Adding spatial proxies to the baseline naive model with only a DEM resulted in different patterns and smoother predicted surfaces (Fig. 8). Comparing naive models with spatial proxies and complete models, spatial patterns were quite similar but more local variation could be appreciated in the latter. Differences between maps derived from complete models with and without proxies were minor."

"Feature extrapolation was the highest in naive models, where proxies had a larger importance that translated into mapping artefacts that were especially evident in the coordinates model (Fig. 9). Unlike the temperature

case study, the predicted surfaces of naive models with proxies and complete models were very different, suggesting that the added geographical predictors could not successfully account for the missing predictors. Prediction maps for complete models with different spatial proxies were much more similar."

*Table 2 and 3: Where are the standard errors on these estimates? (Yes, I understood that some of them are a bit a pain to compute for one of the models. Still, without an estimate of the error, how can the reader interpret a value of "0.92" vs "0.87"? Might well be the same value if SD=0.4.)*

We agree with the reviewer on the importance of showing the standard deviation of the statistics to be able to ascertain whether the differences are relevant. In the revised version of the manuscript, we have computed these for random k-fold cross-validation. For kNNDM, however, we computed statistics by stacking all out-of-sample predicted and observed values, so these could not be calculated (lines 169-171):

"Estimation of RMSE was done globally to account for the different fold sizes in kNNDM (Linnenbrink et al. 2023), i.e. we stacked all predictions in the different folds and computed the RMSE from all samples simultaneously, rather than computing the RMSE within each fold and then averaging."

*I missed the discussion of some existing approaches to manage space into ML.*

*Hajjem, A., Bellavance, F., & Larocque, D. (2011). Mixed effects regression trees for clustered data. Statistics & Probability Letters, 81(4), 451–459. https://doi.org/10.1016/j.spl.2010.12.003*

*Hajjem, A., Bellavance, F., & Larocque, D. (2014). Mixed-effects random forest for clustered data. Journal of Statistical Computation and Simulation, 84(6), 1313–1328. https://doi.org/10.1080/00949655.2012.741599*

*Li, L., Girguis, M., Lurmann, F., Wu, J., Urman, R., Rappaport, E., Ritz, B., Franklin, M., Breton, C., Gilliland, F., & Habre, R. (2019). Cluster-based bagging of constrained mixed-effects models for high spatiotemporal resolution nitrogen oxides prediction over large regions. Environment International, 128, 310–323. https://doi.org/10.1016/j.envint.2019.04.057*

*Li, L., Lurmann, F., Habre, R., Urman, R., Rappaport, E., Ritz, B., Chen, J.-C., Gilliland, F. D., & Wu, J. (2017). Constrained mixed-effect models with ensemble learning for prediction of nitrogen oxides concentrations at high spatiotemporal resolution. Environmental Science & Technology, 51(17), 9920–9929. https://doi.org/10.1021/acs.est.7b01864*

*Zhan, Y., Luo, Y., Deng, X., Chen, H., Grieneisen, M. L., Shen, X., Zhu, L., & Zhang, M. (2017). Spatiotemporal prediction of continuous daily PM2.5 concentrations across China using a spatially explicit machine learning algorithm. Atmospheric Environment, 155, 129–139. https://doi.org/10.1016/j.atmosenv.2017.02.023*

*Kattenborn, T., Schiefer, F., Frey, J., Feilhauer, H., Mahecha, M. D., & Dormann, C. F. (2022). Spatially autocorrelated training and validation samples inflate performance assessment of convolutional neural networks. ISPRS Open Journal of Photogrammetry and Remote Sensing, 5, 100018. https://doi.org/10.1016/j.ophoto.2022.100018*

*Le Rest, K., Pinaud, D., Monestiez, P., Chadoeuf, J., & Bretagnolle, V. (2014). Spatial leave-one-out cross-validation for variable selection in the presence of spatial autocorrelation. Global Ecology and Biogeography, 23, 811–820. https://doi.org/10.1111/geb.12161*

*Ploton, P., Mortier, F., Réjou-Méchain, M., Barbier, N., Picard, N., Rossi, V., Dormann, C., Cornu, G., Viennois, G., Bayol, N., Lyapustin, A., Gourlet-Fleury, S., & Pélissier, R. (2020). Spatial validation reveals poor predictive performance of large-scale ecological mapping models. Nature Communications, 11(1), Article 1. https://doi.org/10.1038/s41467-020-18321-y*

In the introduction we now devote a paragraph to alternative modelling approaches, where these references have been included (lines 106-109):

"As an alternative to modelling with spatial proxies, other methods that *do* involve algorithmic modifications have been proposed, including mixed effects tree-based models that account for correlated data (Hajjem et al. 2011, 2014), spatially-aware resampling methods (Li et al. 2019), as well as geographically weighted ML algorithms (Georganos et al. 2021, Zhan et al. 2017)."

Validation references have been included in the validation and model selection paragraph of the introduction (lines 99-103):

"Several spatial CV methods have been proposed to address the limitations of standard validation approaches (Roberts et al. 2017, Ploton et al. 2020, Kattenborn et al. 2022) using CV based on spatial blocking (Wenger & Olden 2012, Valavi et al. 2019), buffering (Telford & Birks 2009, Le Rest et al. 2014), clustering (Wang et al. 2023), as well as sampling-intensity weighted CV and model-based geostatistical approaches (de Bruin et al. 2022)."

**References**

Behrens, T. & Viscarra Rossel, R. A. (2020), 'On the interpretability of predictors in spatial data science: The information horizon', *Scientific Reports* **10**(1), 16737.

Breiman, L. (2001), 'Random forests', *Machine learning* **45**(1), 5–32.

de Bruin, S., Brus, D. J., Heuvelink, G. B., van Ebbenhorst Tengbergen, T. & Wadoux, A. M.-C. (2022), 'Dealing with clustered samples for assessing map accuracy by cross-validation', *Ecological Informatics* **69**, 101665.
**URL:** *https://linkinghub.elsevier.com/retrieve/pii/S1574954122001145*

F. Dormann, C., M. McPherson, J., B. Araújo, M., Bivand, R., Bolliger, J., Carl, G., G. Davies, R., Hirzel, A., Jetz, W., Daniel Kissling, W., Kühn, I., Ohlemüller, R., R. Peres-Neto, P., Reineking, B., Schröder, B., M. Schurr, F. & Wilson, R. (2007), 'Methods to account for spatial autocorrelation in the analysis of species distributional data: a review', *Ecography* **30**(5), 609–628.
**URL:** *https://nsojournals.onlinelibrary.wiley.com/doi/abs/10.1111/j.2007.0906-7590.05171.x*

Fourcade, Y., Besnard, A. G. & Secondi, J. (2018), 'Paintings predict the distribution of species, or the challenge of selecting environmental predictors and evaluation statistics', *Global Ecology and Biogeography* **27**(2), 245–256.
**URL:** *https://onlinelibrary.wiley.com/doi/abs/10.1111/geb.12684*

Georganos, S., Grippa, T., Gadiaga, A. N., Linard, C., Lennert, M., Vanhuysse, S., Mboga, N., Wolff, E. & Kalogirou, S. (2021), 'Geographical random forests: a spatial extension of the random forest algorithm to address spatial heterogeneity in remote sensing and population modelling', *Geocarto International* **36**(2), 121–136.
**URL:** *https://doi.org/10.1080/10106049.2019.1595177*

Hajjem, A., Bellavance, F. & Larocque, D. (2011), 'Mixed effects regression trees for clustered data', *Statistics & Probability Letters* **81**(4), 451–459.
**URL:** *https://www.sciencedirect.com/science/article/pii/S0167715210003433*

Hajjem, A., Bellavance, F. & Larocque, D. (2014), 'Mixed-effects random forest for clustered data', *Journal of Statistical Computation and Simulation* **84**(6), 1313–1328.
**URL:** *https://doi.org/10.1080/00949655.2012.741599*

Hengl, T. (2007), 'A practical guide to geostatistical mapping of environmental variables.', *Office for Official Publications of the European Communities* .
**URL:** *https://publications.jrc.ec.europa.eu/repository/handle/JRC38153*

Hengl, T., Nussbaum, M., Wright, M. N., Heuvelink, G. B. & Gräler, B. (2018), 'Random forest as a generic framework for predictive modeling of spatial and spatio-temporal variables', *PeerJ* **6**, e5518.

Hijmans, R. J. (2022), *terra: Spatial Data Analysis*. R package version 1.6-47.
**URL:** *https://CRAN.R-project.org/package=terra*

Kattenborn, T., Schiefer, F., Frey, J., Feilhauer, H., Mahecha, M. D. & Dormann, C. F. (2022), 'Spatially autocorrelated training and validation samples inflate performance assessment of convolutional neural networks', *ISPRS Open Journal of Photogrammetry and Remote Sensing* **5**, 100018.
**URL:** *https://www.sciencedirect.com/science/article/pii/S2667393222000072*

Kuhn, M. (2022), *caret: Classification and Regression Training*. R package version 6.0-93.
**URL:** *https://CRAN.R-project.org/package=caret*

Kuhn, M. & Johnson, K. (2019), *Feature engineering and selection: A practical approach for predictive models*, Chapman and Hall/CRC.

Le Rest, K., Pinaud, D., Monestiez, P., Chadoeuf, J. & Bretagnolle, V. (2014), 'Spatial leave-one-out cross-validation for variable selection in the presence of spatial autocorrelation', *Global Ecology and Biogeography* **23**(7), 811–820.
**URL:** *https://onlinelibrary.wiley.com/doi/abs/10.1111/geb.12161*

Li, L., Girguis, M., Lurmann, F., Wu, J., Urman, R., Rappaport, E., Ritz, B., Franklin, M., Breton, C., Gilliland, F. & Habre, R. (2019), 'Cluster-based bagging of constrained mixed-effects models for high spatiotemporal resolution nitrogen oxides prediction over large regions', *Environment International* **128**, 310–323.
**URL:** *https://www.sciencedirect.com/science/article/pii/S0160412018331441*

Linnenbrink, J., Milà, C., Ludwig, M. & Meyer, H. (2023), 'knndm: k-fold nearest neighbour distance matching cross-validation for map accuracy estimation', *EGUsphere* **2023**, 1–16.
**URL:** *https://egusphere.copernicus.org/preprints/2023/egusphere-2023-1308/*

Meyer, H., Milà, C., Ludwig, M. & Linnenbrink, J. (2023), *CAST: 'caret' Applications for Spatial-Temporal Models.* https://github.com/HannaMeyer/CAST, https://hannameyer.github.io/CAST/.

Meyer, H. & Pebesma, E. (2021), 'Predicting into unknown space? estimating the area of applicability of spatial prediction models', *Methods in Ecology and Evolution* **12**(9), 1620–1633.

Meyer, H., Reudenbach, C., Wöllauer, S. & Nauss, T. (2019), 'Importance of spatial predictor variable selection in machine learning applications – moving from data reproduction to spatial prediction', *Ecological Modelling* **411**, 108815.
**URL:** *https://www.sciencedirect.com/science/article/pii/S0304380019303230*

Milà, C., Mateu, J., Pebesma, E. & Meyer, H. (2022), 'Nearest neighbour distance matching leave-one-out cross-validation for map validation', *Methods in Ecology and Evolution* **13**(6), 1304–1316.
**URL:** *https://besjournals.onlinelibrary.wiley.com/doi/abs/10.1111/2041-210X.13851*

Pebesma, E. (2018), 'Simple Features for R: Standardized Support for Spatial Vector Data', *The R Journal* **10**(1), 439–446.
**URL:** *https://doi.org/10.32614/RJ-2018-009*

Pebesma, E. J. (2004), 'Multivariable geostatistics in s: the gstat package', *Computers & Geosciences* **30**(7), 683 – 691.
**URL:** *http://www.sciencedirect.com/science/article/pii/S0098300404000676*

Ploton, P., Mortier, F., Réjou-Méchain, M., Barbier, N., Picard, N., Rossi, V., Dormann, C., Cornu, G., Viennois, G., Bayol, N., Lyapustin, A., Gourlet-Fleury, S. & Pélissier, R. (2020), 'Spatial validation reveals poor predictive performance of large-scale ecological mapping models', **11**(1), 4540.
**URL:** *https://www.nature.com/articles/s41467-020-18321-y*

R Core Team (2022), *R: A Language and Environment for Statistical Computing*, R Foundation for Statistical Computing, Vienna, Austria.
**URL:** *https://www.R-project.org/*

Roberts, D. R., Bahn, V., Ciuti, S., Boyce, M. S., Elith, J., Guillera-Arroita, G., Hauenstein, S., Lahoz-Monfort, J. J., Schröder, B., Thuiller, W. et al. (2017), 'Cross-validation strategies for data with temporal, spatial, hierarchical, or phylogenetic structure', *Ecography* **40**(8), 913–929.

Saha, A., Basu, S. & Datta, A. (2022), *RandomForestsGLS: Random Forests for Dependent Data.* R package version 0.1.4.
**URL:** *https://CRAN.R-project.org/package=RandomForestsGLS*

Saha, A., Basu, S. & Datta, A. (2023), 'Random forests for spatially dependent data', *Journal of the American Statistical Association* **118**(541), 665–683.
**URL:** *https://doi.org/10.1080/01621459.2021.1950003*

Sekulić, A., Kilibarda, M., Heuvelink, G. B., Nikolić, M. & Bajat, B. (2020), 'Random forest spatial interpolation', *Remote Sensing* **12**(10).
**URL:** *https://www.mdpi.com/2072-4292/12/10/1687*

Telford, R. & Birks, H. (2009), 'Evaluation of transfer functions in spatially structured environments', *Quaternary Science Reviews* **28**(13), 1309 – 1316.

Tennekes, M. (2018), 'tmap: Thematic maps in R', *Journal of Statistical Software* **84**(6), 1–39.

Valavi, R., Elith, J., Lahoz-Monfort, J. J. & Guillera-Arroita, G. (2019), 'blockcv: An r package for generating spatially or environmentally separated folds for k-fold cross-validation of species distribution models', *Methods in Ecology and Evolution* **10**(2), 225–232.

Wadoux, A. M.-C., Heuvelink, G. B., de Bruin, S. & Brus, D. J. (2021), 'Spatial cross-validation is not the right way to evaluate map accuracy', *Ecological Modelling* **457**, 109692.
**URL:** *https://www.sciencedirect.com/science/article/pii/S0304380021002489*

Wadoux, A. M.-C., Minasny, B. & McBratney, A. B. (2020), 'Machine learning for digital soil mapping: Applications, challenges and suggested solutions', *Earth-Science Reviews* **210**, 103359.
**URL:** *https://www.sciencedirect.com/science/article/pii/S0012825220304050*

Wadoux, A. M. J.-C., Samuel-Rosa, A., Poggio, L. & Mulder, V. L. (2020), 'A note on knowledge discovery and machine learning in digital soil mapping', *European Journal of Soil Science* **71**(2), 133–136.
**URL:** *https://bsssjournals.onlinelibrary.wiley.com/doi/abs/10.1111/ejss.12909*

Wang, Y., Khodadadzadeh, M. & Zurita-Milla, R. (2023), 'Spatial+: A new cross-validation method to evaluate geospatial machine learning models', *International Journal of Applied Earth Observation and Geoinformation* **121**, 103364.
**URL:** *https://www.sciencedirect.com/science/article/pii/S1569843223001887*

Wenger, S. J. & Olden, J. D. (2012), 'Assessing transferability of ecological models: an underappreciated aspect of statistical validation', *Methods in Ecology and Evolution* **3**(2), 260–267.

Wickham, H. (2016), *ggplot2: Elegant Graphics for Data Analysis*, Springer-Verlag New York.
**URL:** *https://ggplot2.tidyverse.org*

Wright, M. N. & Ziegler, A. (2017), 'ranger: A fast implementation of random forests for high dimensional data in C++ and R', *Journal of Statistical Software* **77**(1), 1–17.

Wylie, B. K., Pastick, N. J., Picotte, J. J. & Deering, C. A. (2019), 'Geospatial data mining for digital raster mapping', *GIScience & Remote Sensing* **56**(3), 406–429.
**URL:** *https://www.tandfonline.com/doi/abs/10.1080/15481603.2018.1517445*

Zhan, Y., Luo, Y., Deng, X., Chen, H., Grieneisen, M. L., Shen, X., Zhu, L. & Zhang, M. (2017), 'Spatiotemporal prediction of continuous daily pm2.5 concentrations across china using a spatially explicit machine learning algorithm', *Atmospheric Environment* **155**, 129–139.
**URL:** *https://www.sciencedirect.com/science/article/pii/S1352231017300936*